# Statistical Inference for Generative Model Comparison

**Zijun Gao**[*]                                                                              *zijungao@marshall.usc.edu*
*Marshall School of Business, University of Southern California*

**Han Su**[*]                                                                                        *suhan@mail.bnu.edu.cn*
*School of Statistics, Beijing Normal University*

**Yan Sun**[*]                                                                                              *yan.sun@njit.edu*
*Department of Mathematical Science, New Jersey Institute of Technology*

**Reviewed on OpenReview:** *https://openreview.net/forum?id=PXL6SBxh0q*

## Abstract

Generative models have achieved remarkable success across a range of applications, yet their evaluation still lacks principled uncertainty quantification. In this paper, we develop a method for comparing how close different generative models are to the underlying distribution of test samples. Particularly, our approach employs the Kullback-Leibler (KL) divergence to measure the distance between a generative model and the unknown test distribution, as KL requires no tuning parameters such as the kernels used by RKHS-based distances. And the relative KL divergence is the only $f$-divergence that admits a crucial cancellation of the hard-to-estimate term to enable the faithful uncertainty quantification. Furthermore, we extend our method to comparing conditional generative models and leverage Edgeworth expansions to address limited-data settings. On simulated datasets with known ground truth, we show that our approach realizes effective coverage rates, and has higher power compared to kernel-based methods. When applied to generative models on image and text datasets, our procedure yields conclusions consistent with benchmark metrics but with statistical confidence. The source code to reproduce our experiments is available at https://github.com/sylydya/compare-generative-models.

**Keywords:** Uncertainty quantification, Generative model evaluation, Edgeworth expansion, $f$-divergence

## 1 Introduction

Generative models have achieved remarkable success across a wide range of applications, demonstrating their versatility in areas such as image synthesis, natural language processing, and scientific discovery (Achiam et al., 2023; Goodfellow et al., 2014; Van Den Oord et al., 2016a; Karras et al., 2020). As the number of generative models continues to grow rapidly, practitioners increasingly face the challenge of selecting the most appropriate model. This underscores the need for principled methods of performance evaluation and model comparison.

Despite the advances in model development, approaches for assessing and comparing generative models remain relatively underexplored. One class of evaluation methods rely on human judgment, which are inherently subjective and difficult to scale. An alternative line of methods use standardized benchmark tests (Liu et al., 2024; Gallifant et al., 2024) (such as AIME 2024 and MATH 500), which are limited to output types for which such benchmark tests exist. In addition, various quantitative metrics have been proposed for different types of generated outputs, such as Wasserstein distance-based metric (e.g., Fréchet Inception Distance, FID) for images and perplexity for texts. However, these metrics lack uncertainty quantification,

---

[*]Alphabetical order

Figure 1: An example of our method applied to comparing the diffusion models (DDIMs with different numbers of denoising steps $S$). Here, $\mathbb{P}$ represents the distribution of the test images, $\hat{\mathbb{P}}_1$ corresponds to the DDIM model with $S = 50$ denoising steps, and $\hat{\mathbb{P}}_2$ corresponds to the DDIM model with $S = 100$ denoising steps. Our method demonstrates that the confidence interval for the relative score $\delta(\hat{\mathbb{P}}_1, \hat{\mathbb{P}}_2)$ is significantly negative, indicating that $\hat{\mathbb{P}}_2$ with $S = 100$ achieves significantly better performance. While consistent with the FID reported in Song et al. (2021), our method further quantifies the statistical significance of the performance difference, which FID cannot provide.

making it difficult to determine whether observed differences reflect true performance gaps or arise solely from variability of the test sample[1].

The lack of satisfactory model assessment methods can be partially attributed to the challenge that evaluating generative models requires comparing output distributions rather than individual data points. For predictors or classifiers, the performance can be directly measured by comparing the generated outputs to the associated true labels. In contrast, the quality of a generative model is determined by how closely the *distribution* of its generated data matches that of the input data, rather than the similarity between generated data points and input data points (known as the reconstruction error). Moreover, generative models often produce high-dimensional outputs[2], making the distribution comparison even more challenging.

In this paper, we develop methods for comparing generative models with uncertainty quantification. Specifically, we make the following contributions:

- We propose a method for evaluating and comparing generative models based on the *relative* score of KL divergence. Focusing on the relative score enables the cancellation of a hard-to-estimate quantity in the absolute score, leading to favorable statistical properties.

- We develop an unbiased estimator in the form of a first-order U-statistic that achieves convergence at the parametric rate. We explicitly characterize the asymptotic distribution of our estimator. To improve the finite sample performance of our method, we refine the limiting distribution analysis via Edgeworth expansion, which leads to faithful coverage even with a very small sample size.

- We further extend our framework from unconditional to *conditional* generative models, which are central in practice (e.g., text-to-image models, language models). This allows principled statistical comparison of conditional distributions.

- On simulated datasets with known ground truth, we demonstrate that our approach constructs faithful confidence intervals and achieves higher power compared to baselines. Furthermore, we demonstrate the effectiveness of our method in evaluating generative models on the CIFAR-10 dataset and assessing large language models (LLM) on Wikitext-2 and TriviaQA data sets.

---

[1]In practice, metrics such as FID are usually computed given a dataset for evaluation. In our work, we view the ground truth distribution as an unknown population distribution, and the dataset for evaluation is viewed as a finite sample from the distribution. This finite-sample approximation is the main source of the uncertainty that we quantify in our work. From this perspective, FID is viewed as the Wasserstein / Fréchet distance between the transformed distribution induced by the Inception model.

[2]The output is typically ultra-high-dimensional, for example, a 1080p resolution image consists of around $2 \times 10^6$ pixels with (approximately) continuous value. Particularly, an image is represented as a matrix of size $d_1 \times d_2$, where $d = d_1 \cdot d_2$ is the total number of pixels. Each pixel consists of three color channels (e.g., RGB), where each channel takes integer values ranging from 0 to 255. As a result, a single pixel can represent $256^3 \approx 16.7 \times 10^6$ possible color combinations. The pixel values can effectively be regarded as continuous.

**Organization**. In Section 2, we formulate the problem of evaluating generative models. In Section 3, we introduce the concept of relative score which enables a key cancellation, and propose our uncertainty quantification method. In Section 4, we extend the method to conditional generative model comparison and the cases with limited test data. In Section 5, we evaluate the numerical performance of our methods using both simulated and real image and text datasets. All proofs and additional literature, method details, and empirical results are provided in the Appendix.

**Notations**. Let $\Omega$ denote the space of test data, and let $\mathbb{P}$ represent the true target distribution. Let $\hat{\mathbb{P}}_1$ denote the distribution of data generated by a generative model. If the generative process of the generative model involves sampling a random noise vector, we use $d_1$ to denote its dimension. Similarly we define $\hat{\mathbb{P}}_2$ and $d_2$. Let $\mathbb{P}_n$ represent the empirical distribution of $n$ observations from $\mathbb{P}$, and similarly for $\hat{\mathbb{P}}_{1,n}$, $\hat{\mathbb{P}}_{2,n}$. We denote densities by lower-case letters, e.g., $p$ as the density of $\mathbb{P}$. We use $\phi_d$ to denote the density of the standard multivariate Gaussian density in $\mathbb{R}^d$. For $x \in \mathbb{R}^d$, we use $\|x\|$ to denote its $\ell_2$ norm.

## 2 Formulation and background

### 2.1 Comparison of generative models

In this paper, we focus on the case where a set of $n_{\text{test}}$ test data points $Y_i$, $1 \leq i \leq n_{\text{test}}$ independently and identically distributed (i.i.d.) from the target distribution $\mathbb{P}$, are provided. We aim to compare generative models using the test dataset. For conciseness, we focus on the comparison of two models, denoted by $\hat{\mathbb{P}}_1$ and $\hat{\mathbb{P}}_2$, while the generalization to multiple-model comparisons is straightforward.

Generative models are designed to produce samples whose distribution approximates the true underlying distribution $\mathbb{P}$. To evaluate a generative model $\hat{\mathbb{P}}_1$ quantitatively, a dissimilarity metric between $\mathbb{P}$ and $\hat{\mathbb{P}}_1$ is computed. In this work, we use the KL divergence, also known as relative entropy, for the following reasons: (1) KL divergence is the only $f$-divergence that satisfies a cancellation property (see Proposition 1), which enables unbiased estimation and valid uncertainty quantification; (2) KL divergence strongly penalizes mismatches in regions where one distribution assigns probability mass and the other does not, desirable when rare but consequential events shall drive conclusions; (3) unlike kernel or representation-learning–based metrics, KL divergence requires neither choosing a kernel nor learning latent features, reducing the risk of uninformative comparisons caused by suboptimal hyperparameter choice or insufficient tuning.

**Assumption 2.1.** *Suppose the test data distribution $\mathbb{P}$ and the generated data distribution $\hat{\mathbb{P}}_1$ admit densities, and $\mathbb{P}$ is absolutely continuous with respect to $\hat{\mathbb{P}}_1$.*

Under Assumption 2.1, we formally define the absolute score for the generative model[3] associated with $\hat{\mathbb{P}}_1$ as the negative KL divergence between $\mathbb{P}$ and $\hat{\mathbb{P}}_1$,

$$s(\hat{\mathbb{P}}_1) := -\text{KL}(\mathbb{P}\|\hat{\mathbb{P}}_1) = -\int \log\left(\frac{p(y)}{\hat{p}_1(y)}\right) p(y)\, dy, \tag{1}$$

where we denote the density of $\mathbb{P}$, $\hat{\mathbb{P}}_1$ by $p$, $\hat{p}_1$, respectively. A larger absolute score $s(\hat{\mathbb{P}}_1)$ indicates better performance of the generative model $\hat{\mathbb{P}}_1$.

In this paper, we focus on comparing generative models whose sampling density evaluated at a test data point is accessible. Many existing models satisfy this property. Details about the computation of the density function for different generative models are provided in Section C:

- **Generative models for image**. A significant proportion of generative models for images, including variational auto-encoders (Kingma, 2013), autoregressive models (Van Den Oord et al., 2016b), normalizing flows (Rezende & Mohamed, 2015; Dinh et al., 2016), diffusion models (Ho et al., 2020; Song & Ermon, 2019; Song et al., 2021), consist of a forward and a reverse process. Any image generative model that admits an accessible and invertible reverse process, such as normalizing flows

---

[3]For conciseness, we use the generating distribution to represent its corresponding generative model. For example, we may refer to the generative model as $\hat{\mathbb{P}}_1$.

(Chen et al., 2018), Denoising Diffusion Implicit Model (DDIM) (Song et al., 2021), permits the density evaluation at a test data point.

- **Generative models for text**. Autoregressive language models, which generate text token by token conditioned on previous context, are the most widely used LLMs nowadays. Let $\mathcal{V}$ denote the vocabulary. Let $y = (r_1, r_2, \ldots, r_L)$, $r_i \in \mathcal{V}$ denote a response sequence, and let $r_{1:i}$ denote the first to the $i$-th tokens of $r$. Let $\mathbb{P}$ denote the ground truth probability of responses. An autoregressive language model defines the probability of the next token given previous tokens as $\hat{\mathbb{P}}_1(r_{i+1}|r_{1:i})$. By the chain rule of probability, the joint probability of the sequence $r$ is

$$\hat{\mathbb{P}}_1(r) = \prod_{i=0}^{L-1} \hat{\mathbb{P}}_1(r_{i+1}|r_{1:i}),$$

where $\hat{\mathbb{P}}_1(r_i|r_{1:0}) = \hat{\mathbb{P}}_1(r_1)$. For open-source LLMs, the estimated probability $\hat{p}_1(r)$ at a test data point is typically accessible through the forward pass.

## 2.2 Related works

We review literature on generative model evaluation based on distributional closeness. We acknowledge that generative models are also evaluated along other dimensions (e.g., overfitting, diversity, fidelity, memorization), which are important in practice but lie beyond the scope of this work.

**Wasserstein-distance-based evaluation** Wasserstein-$p$ distance (Villani et al., 2009) is defined as

$$W_p(\mathbb{P}, \hat{\mathbb{P}}_1) = \inf_{\gamma \in \Gamma(\mathbb{P}, \hat{\mathbb{P}}_1)} \mathbb{E}^{1/p}_{(x,y)\sim\gamma} \left[ \|x - y\|_p^p \right],$$

where $\Gamma(\mathbb{P}, \hat{\mathbb{P}}_1)$ is the set of couplings between $\mathbb{P}$ and $\hat{\mathbb{P}}_1$. There are several challenges of performing inference for evaluation methods based on the Wasserstein-$p$ distance. First, Del Barrio & Loubes (2019) shows that[4]

$$\sqrt{n}\left( W_2^2(\mathbb{P}_n, \hat{\mathbb{P}}_{1,n}) - \mathbb{E}[W_2^2(\mathbb{P}_n, \hat{\mathbb{P}}_{1,n})] \right) \xrightarrow{d} \mathcal{N}\left( 0, \sigma^2(\mathbb{P}, \hat{\mathbb{P}}_1) \right),$$

where the asymptotic variance $\sigma^2(\mathbb{P}, \hat{\mathbb{P}}_1)$ can be estimated consistently using a plug-in estimator. The issue is that the center $\mathbb{E}[W_2^2(\mathbb{P}_n, \hat{\mathbb{P}}_{1,n})]$ is different from the desired $W_2^2(\mathbb{P}, \hat{\mathbb{P}}_1)$, and the gap between $\mathbb{E}[W_2^2(\mathbb{P}_n, \hat{\mathbb{P}}_{1,n})]$ and $W_2^2(\mathbb{P}, \hat{\mathbb{P}}_1)$ scales as $n^{-1/d}$ Villani et al. (2009). Second, for the relative Wasserstein-2 distance $W_2^2(\mathbb{P}, \hat{\mathbb{P}}_1)$, the joint asymptotic distribution of $(W_2^2(\mathbb{P}_n, \hat{\mathbb{P}}_{1,n}), W_2^2(\mathbb{P}_n, \hat{\mathbb{P}}_{2,n}))$ is required. It remains unclear whether $W_2^2(\mathbb{P}_n, \hat{\mathbb{P}}_{1,n})$ and $W_2^2(\mathbb{P}_n, \hat{\mathbb{P}}_{2,n})$ are asymptotically jointly Gaussian, as well as what the exact form of their covariance matrix is[5]. Third, subsampling methods Dümbgen (1993) can be employed for conducting inference for Wasserstein-p distances; however, subsampling is computationally expensive, especially given that the Wasserstein-p distance is already difficult to compute. For a comprehensive review of these issues, see Panaretos & Zemel (2019).

**FID/IS-based evaluation** We review the inception score (IS) (Salimans et al., 2016) and the Frechet Inception Distance (FID) (Heusel et al., 2017; Jayasumana et al., 2024; Jiralerspong et al., 2023), two most commonly used quantitative scores for generative models. IS evaluates the quality of a generative model by applying a separate, pretrained image classification model to a batch of images generated by the model. IS is maximized when the classifier confidently predicts a single label for each image, or when the predictions are evenly distributed across all possible labels. The quality of IS depends heavily on the quality of the classifier (if the classifier consistently outputs a single label for all images, the IS becomes uninformative). Another disadvantage is that IS does not compare generated images to test images. FID compares the distribution between the distribution of test images and that of generated images. Mathematically, FID aims to approximate the Wasserstein-2 distance between the two distribution in two steps: (1) mapping the

---

[4]CLT results of general Wasserstein-$p$ distance are largely unknown.

[5]The off-diagonal values of the covariance matrix is non-zero because both $W_2^2(\mathbb{P}_n, \hat{\mathbb{P}}_{1,n})$ and $W_2^2(\mathbb{P}_n, \hat{\mathbb{P}}_{2,n})$ depend on $\mathbb{P}_n$.

real and generated images to $\mathbb{R}^d$ separately by passing them through the final layer of an image classifier to extract essential features; (2) fitting multivariate Gaussian distributions to the transformed data in $\mathbb{R}^d$ and computing the Wasserstein-2 distance between these multivariate Gaussians. The approximation accuracy depends on how well the transformation to $\mathbb{R}^d$ captures the data characteristics and the quality of the multivariate Gaussians fit to the transformed data (the covariance matrix is typically not diagonal and the estimation involves $O(d^2)$ elements). Chong & Forsyth (2020) shows that FID could be significantly biased in finite sample.

**Kernel-based evaluation**  There is a line of kernel-based evaluation metrics which admit uncertainty quantification (Bińkowski et al., 2018; Liu et al., 2016; Chwialkowski et al., 2016; Bounliphone et al., 2016; Kanagawa et al., 2023). However, kernel-based evaluation hinges on having a powerful kernel, which is challenging for complex, high-dimensional data such as images and text. In particular, as we show in the simulations, even carefully designed kernels may fail to capture all fine-grained distinctions. Moreover, kernel-based methods are computationally expensive, requiring quadratic pairwise computations, whereas our method scales linearly in sample size. For kernelized Stein discrepancies (Liu et al., 2016; Kanagawa et al., 2023), scores of generative models are required, which can be expensive to compute. For large language models (LLMs), evaluating the score entails back-propagation through the network, which is substantially more costly than the single forward pass required by our method.

**Likelihood-ratio-based evaluation**  There is a thread of works in using likelihood–ratio for comparing statistical models (Neyman & Pearson, 1933; Wilks, 1938; Vuong, 1989). Given two candidate families of distributions (two statistical models), likelihood–ratio–based tests determines the more suitable family by first fitting each model to the data and then evaluating the difference in the fitted models' log-likelihoods. However, for modern generative models (Song et al., 2021; Dubey et al., 2024), this fitting step is typically computationally expensive or even prohibitive. In contrast, our approach operates on pre-trained generative models and requires no additional fitting. In addition, we complement this line of work by offering conditional comparisons (Section 4), which are important in scenarios where poor performance within specific subpopulations is especially consequential or dangerous. Finally, while this literature adopts log-likelihood to maximize test power, we work with the KL divergence mostly because it is the only $f$-divergence that admits the cancellation property (Proposition 1), crucial for valid inference.

## 3   Relative score for generative model comparison

Despite the popularity of the KL-divergence (our absolute score), its estimation and inference are considerably challenging. According to Zhao & Lai (2020), the minimax optimal rate for estimating the KL divergence between two densities in $\mathbb{R}^d$, based on a sample of size $n_{\text{test}}$ from each density, scales as slow as $n_{\text{test}}^{-2/d}$. In addition, the asymptotic distribution of the KL-divergence is typically intractable except for special cases (Belov & Armstrong, 2011), and computationally-heavy bootstrap or subsampling methods are called for to conduct inference based on the KL-divergence (Arizono & Ohta, 1989).

Instead of investigating the absolute scores of two generative models separately, we propose to directly study the *relative* score, the difference between their absolute scores, defined as

$$\delta(\hat{\mathbb{P}}_1, \hat{\mathbb{P}}_2) = -\text{KL}(\mathbb{P}\|\hat{\mathbb{P}}_1) + \text{KL}(\mathbb{P}\|\hat{\mathbb{P}}_2). \tag{2}$$

The relative score aims to quantify the performance gap between the two generative models. If $\delta(\hat{\mathbb{P}}_1, \hat{\mathbb{P}}_2) > 0$, it implies that $\text{KL}(\mathbb{P}\|\hat{\mathbb{P}}_1) < \text{KL}(\mathbb{P}\|\hat{\mathbb{P}}_2)$, and we conclude that $\hat{\mathbb{P}}_1$ is superior to $\hat{\mathbb{P}}_2$.

In contrast to the absolute score, the relative score benefits from a nice cancellation of some hard-to-estimate term, facilitating its estimation and inference. Explicitly, the absolute score (1) contains two terms, with $\int \log(\log(p(y)) \, d\mathbb{P}(y)$ being less tractable than $\int \hat{p}_1(y)) \, d\mathbb{P}(y)$, as the generating mechanism $\hat{p}_1(y)$ is essentially known, i.e.,

$$s(\hat{\mathbb{P}}_1) = -\Big( \underbrace{\int \log\left(p(y)\right) \, d\mathbb{P}(y)}_{\text{Challenging}} - \underbrace{\int \log\left(\hat{p}_1(y)\right) \, d\mathbb{P}(y)}_{\text{Tractable}} \Big).$$

By the definition (2), $\delta(\hat{\mathbb{P}}_1, \hat{\mathbb{P}}_2)$ equals

$$
-\left( \underbrace{\int \log(p(y))\, d\mathbb{P}(y)}_{\text{Challenging}} - \underbrace{\int \log(\hat{p}_1(y))\, d\mathbb{P}(y)}_{\text{Tractable 1}} \right) + \left( \underbrace{\int \log(p(y))\, d\mathbb{P}(y)}_{\text{Challenging}} - \underbrace{\int \log(\hat{p}_2(y))\, d\mathbb{P}(y)}_{\text{Tractable 2}} \right)
$$

$$
= \underbrace{\int \log(\hat{p}_1(y))\, d\mathbb{P}(y)}_{\text{Tractable 1}} - \underbrace{\int \log(\hat{p}_2(y))\, d\mathbb{P}(y)}_{\text{Tractable 2}}.
\tag{3}
$$

Here the challenging term $\int \log(p(y))\, d\mathbb{P}(y)$, appearing in both $\mathrm{KL}(\mathbb{P}\|\hat{\mathbb{P}}_1)$ and $\mathrm{KL}(\mathbb{P}\|\hat{\mathbb{P}}_2)$, cancels out. The remaining term $\int \log(\hat{p}_1(y)) - \log(\hat{p}_2(y))\, d\mathbb{P}(y)$ is the expectation of an effectively known function $\log(\hat{p}_1(y)) - \log(\hat{p}_2(y))$ regarding the test data distribution. Therefore, the relative score can be efficiently estimated using a first-order U-statistic based on a set of test data points, detailed in Section 3.1 below.

We conclude this section by showing that the attractive cancellation (3) in the relative score is *unique* to our choice of KL divergence. For a convex function $f : [0, +\infty) \to (-\infty, +\infty]$ such that $f(x)$ is finite for all $x > 0$, $f(1) = 0$, and $f(0) = \lim_{t\to 0^+} f(t)$, the f-divergence of $\mathbb{P}$ from $\hat{\mathbb{P}}_1$ is defined as $D_f(\mathbb{P}\|\hat{\mathbb{P}}_1) := \int_\Omega f\left( \frac{p(y)}{\hat{p}_1(y)} \right) \hat{p}_1(y) dy$. KL-divergence is a special case of $f$-divergence with $f(x) = x \log(x)$.

**Proposition 1.** *For an $f$-divergence with $f \in C^1$, if there exists a function $g$ such that for any $\hat{\mathbb{P}}_1$, $\hat{\mathbb{P}}_2$, $\mathbb{P}$,*

$$
D_f(\mathbb{P}\|\hat{\mathbb{P}}_1) - D_f(\mathbb{P}\|\hat{\mathbb{P}}_2) = \int g(\hat{\mathbb{P}}_1, \hat{\mathbb{P}}_2) d\mathbb{P},
\tag{4}
$$

*then there exists $\beta \geq 0$ such that $f(x) = \beta x \log(x)$, i.e., $D_f(\mathbb{P}\|\hat{\mathbb{P}}_1) = \beta \mathrm{KL}(\mathbb{P}\|\hat{\mathbb{P}}_1)$.*

We prove Proposition 1 by (1) reducing it to the Cauchy functional equation problem through multiple rounds of re-parametrization; (2) applying the uniqueness of the solution to the Cauchy functional equation. Details are provided in Section B of the Appendix.

## 3.1 Estimation and Inference of Relative Score

### 3.1.1 Estimator

By (3), for generative models with accessible $\hat{p}_1(Y_i)$ and $\hat{p}_2(Y_i)$, we estimate the relative score by the first-order U-statistic (sample mean) on the test dataset,

$$
\hat{\delta}(\hat{\mathbb{P}}_1, \hat{\mathbb{P}}_2) := \frac{1}{n_{\text{test}}} \sum_{i=1}^{n_{\text{test}}} \log(\hat{p}_1(Y_i)) - \log(\hat{p}_2(Y_i)).
\tag{5}
$$

According to the standard property of U-statistics, we establish the following unbiasedness result.

**Proposition 2.** *The estimator $\hat{\delta}(\hat{\mathbb{P}}_1, \hat{\mathbb{P}}_2)$ in (5) is unbiased, i.e., $\mathbb{E}\left[ \hat{\delta}(\hat{\mathbb{P}}_1, \hat{\mathbb{P}}_2) \right] = \delta(\hat{\mathbb{P}}_1, \hat{\mathbb{P}}_2)$.*

In Section C of the Appendix, we detail the computation of our estimator for both image and text generative models discussed in Section 2, along with several acceleration techniques to improve the computational efficiency.

### 3.1.2 Inference

We describe the asymptotic distribution of the estimator in (5) in the following theorem.

**Theorem 3.1.** *Let $V := \mathsf{Var}\left( \log(\hat{p}_1(Y_i)) - \log(\hat{p}_2(Y_i)) \right)$. If $V < \infty$,*

$$
\sqrt{n_{test}} \left( \hat{\delta}(\hat{\mathbb{P}}_1, \hat{\mathbb{P}}_2) - \delta(\hat{\mathbb{P}}_1, \hat{\mathbb{P}}_2) \right) \xrightarrow{d} \mathcal{N}(0, V).
$$

The detailed proof is provided in Section B of the Supplementary Material. Other methods that provide uncertainty quantification (Bounliphone et al., 2016; Liu et al., 2016; Kanagawa et al., 2023) focus on non-KL distances (e.g., kernel-based distances) and establish the joint asymptotic distribution of estimated absolute scores $(\hat{s}(\hat{\mathbb{P}}_1), \hat{s}(\hat{\mathbb{P}}_2))$. In contrast, for the KL divergence in our method, the absolute score estimators $\hat{s}(\hat{\mathbb{P}}_1)$ and $\hat{s}(\hat{\mathbb{P}}_2)$ are heavily biased with intractable asymptotic distribution, but their difference enjoys the cancellation and still converges at the $n^{-1/2}$ rate to a normal limit.

By the law of large numbers, the empirical variance of $\log(\hat{p}_1(Y_i)) - \log(\hat{p}_2(Y_i))$, denoted by $\hat{V}$, is a consistent estimator of $V$. Using Slutsky's theorem (see e.g., Lemma 2.8 of Van der Vaart (2000)), we derive the following corollary of Theorem 3.1.

**Corollary 3.1.** *If $0 < V < \infty$, then*

$$\frac{\hat{\delta}(\hat{\mathbb{P}}_1, \hat{\mathbb{P}}_2) - \delta(\hat{\mathbb{P}}_1, \hat{\mathbb{P}}_2)}{\sqrt{\hat{V}/n_{test}}} \xrightarrow{d} \mathcal{N}(0,1). \tag{6}$$

Corollary 3.1 allows us to perform statistical inference on the relative score, enabling the determination of the better generative model with a specified level of confidence. Explicitly, let $\alpha \in (0,1)$ be the confidence level, and we consider the following confidence interval (CI) of the relative score, denoted by $\widehat{\mathrm{CI}}(\alpha)$,

$$\left[ \hat{\delta}(\hat{\mathbb{P}}_1, \hat{\mathbb{P}}_2) - q_{1-\frac{\alpha}{2}}\sqrt{\frac{\hat{V}}{n_{\text{test}}}}, \ \hat{\delta}(\hat{\mathbb{P}}_1, \hat{\mathbb{P}}_2) + q_{1-\frac{\alpha}{2}}\sqrt{\frac{\hat{V}}{n_{\text{test}}}} \right] \tag{7}$$

where $q_{1-\frac{\alpha}{2}}$ denotes the upper $1 - \alpha/2$ quantile of a standard normal. The following statement states the validity of the confidence interval.

**Corollary 3.2.** *Under the conditions in Corollary 3.1, for any $\alpha \in (0,1)$, $\mathbb{P}\left(\delta(\hat{\mathbb{P}}_1, \hat{\mathbb{P}}_2) \in \widehat{\mathrm{CI}}(\alpha)\right) \to 1 - \alpha$.*

## 4 Extensions

### 4.1 Conditional generative model comparison

Our method can also be extended to the comparison of conditional generative models. Suppose that the data $(X_i, Y_i)_{i=1}^n$ are generated from joint distribution $\mathbb{P}$ with density $p(X, Y)$. Conditional generative models aim to approximate the conditional distribution $p(Y|X)$ via $\hat{p}(Y|X)$. Then $\hat{p}(Y|X)p(X)$ serves as an approximation of the joint distribution[6]. Given two conditional generative models $\hat{\mathbb{P}}_1, \hat{\mathbb{P}}_2$ that aim to approximate the conditional distribution $p(y|x)$ by $\hat{p}_1(y|x)$ and $\hat{p}_2(y|x)$, similar to the Section 3, we consider the absolute score $-\mathrm{KL}(p(x,y)\|\hat{p}_1(y|x)p(x))$. Then the corresponding relative score $\delta(\hat{\mathbb{P}}_1, \hat{\mathbb{P}}_2)$ can be defined as

$$-\mathrm{KL}\left(p(x,y)\|\hat{p}_1(y|x)p(x)\right) + \mathrm{KL}\left(p(x,y)\|\hat{p}_2(y|x)p(x)\right)$$
$$= \int \left(\log \hat{p}_1(y|x) - \log \hat{p}_2(y|x)\right) p(x,y) dy dx,$$

where the hard-to-estimate entropy term also cancels out. Thus, we can define the unbiased estimator by:

$$\hat{\delta}(\hat{\mathbb{P}}_1, \hat{\mathbb{P}}_2) = \frac{1}{n}\sum_{i=1}^n \log \hat{p}_1(y_i|x_i) - \log \hat{p}_2(y_i|x_i), \tag{8}$$

which is again a first-order U-statistic. All theoretical results in Section 3 naturally extend to the conditional setting.

---

[6]In the setting of conditional generative models, the conditioned quantity $X$ is usually given, therefore we don't need to approximate $p(X)$.

## 4.2 Edgeworth expansions for limited test data

Corollary 3.2 ensures asymptotic correct coverage rate of the $\widehat{\mathrm{CI}}(\alpha)$, but when we have limited test data ($n \leq 100$) (which may arise in settings with costly human labels, slow data collection, online experiments with early stopping), or non-Gaussian output (i.e., $\widehat{\mathbb{P}}_i$ deviates from the normal distribution), then $\widehat{\mathrm{CI}}(\alpha)$ may not always provide reliable coverage rate. In this subsection, we focus on further theoretical discussions on the inference of the relative score. By introducing a high-order expansion for the distribution of testing statistics, called Edgeworth Expansions (EEs), we can obtain more accurate statistical inference results. We first present the superiority of EEs over classical Central Limit Theorem (CLT, e.g., Theorem 3.1) with respect to constructing CI of the relative score $\delta(\widehat{\mathbb{P}}_1, \widehat{\mathbb{P}}_2)$ when sample size $n$ is small, as shown in Figure 2. The detailed setting is given in Section 5.1.1.

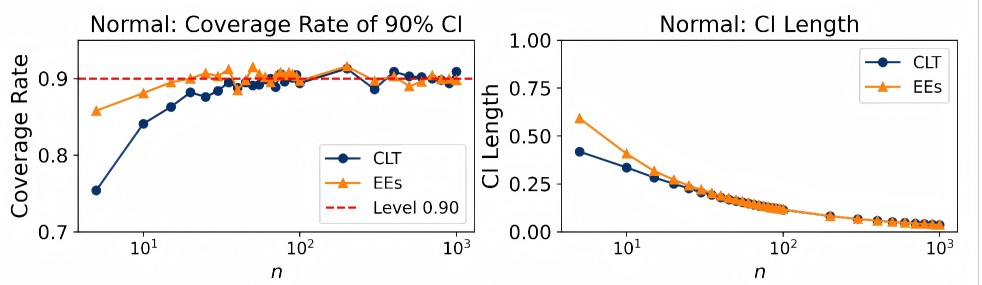

Figure 2: Coverage rates and length of confidence intervals obtained by Edgeworth Expansions (12) and Central Limit Theorem (7) across different sample sizes $n$ using simulated data generated by (13) with $\epsilon = 0.07$.

By utilizing information from higher-order moments, such as skewness and kurtosis, EEs can yield more precise limiting distributions and faster convergence rate against CLT. Specifically, we revisit the proof of Theorem 3.1 of $Z$-statistics $Z_n = (V/n_{\text{test}})^{-1/2}[\hat{\delta}(\widehat{\mathbb{P}}_1, \widehat{\mathbb{P}}_2) - \delta(\widehat{\mathbb{P}}_1, \widehat{\mathbb{P}}_2)]$ in Theorem 3.1. For simplicity, we slightly abuse the notations, let $\{X_i = \log(\hat{p}_1(Y_i)) - \log(\hat{p}_2(Y_i)) \sim F\}_{i=1}^n$ be i.i.d. relative scores, with $\mathbb{E}(X) = \mu, 0 < \mathsf{Var}(X) = \sigma^2 < \infty$, and let $\overline{X}_n = n^{-1} \sum_{i=1}^n X_i$. By the independence of $X_i$, the characteristic function (c.f.) of $Z_n$ has the expansion form based on the first and second moments of $X_i$:

$$\psi_{Z_n}(t) = \psi_{\frac{X_i - \mu}{\sigma}}^n \left( \frac{t}{\sqrt{n}} \right) = \left( 1 - \frac{t^2}{2n} + o \left( \frac{t^2}{n} \right) \right)^n. \tag{9}$$

Combined (9) with the Lévy continuity theorem (see e.g., Theorem 2.13 of Van der Vaart (2000)), we can derive the CLT

$$Z_n := \frac{\sqrt{n}(\overline{X}_n - \mu)}{\sigma} \xrightarrow{d} \mathcal{N}(0, 1). \tag{10}$$

However, in generative models, both $\{(\hat{p}_1(Y_i), \hat{p}_2(Y_i))\}_i$ and $\{X_i\}_i$ are usually far away from Gaussian. CLT unify the Gaussian distribution as the asymptotic distribution for the inference of relative scores $\delta$, which may result in unreliable comparison results, especially when $n$ is relatively small.

In fact, CLT (9) only performs a second-order moment expansion of the c.f. of $Z_n$, and does not utilize higher-order moments such as skewness and kurtosis. Therefore, one can further expand the c.f. of $Z_n$ based on higher-order population moments of $X$ (if exist), such that the limiting distribution of $Z_n$ can be characterized more precisely, further making reliable inferences in generative model comparisons. Based on the calculation of cumulants and the Fourier inversion of $\psi_{Z_n}(t)$, the following theorem gives the EEs of $Z_n$.

**Theorem 4.1.** *Under certain regularity assumptions on the distribution of $X_i$ (details in Section B of the Appendix), the distribution function $F_n(x)$ of $Z_n$ satisfies*

$$F_n(x) = \underbrace{\Phi(x) - n^{-1/2} \frac{\kappa_3}{6} H_2(x)\phi(x) - n^{-1} \left\{ \frac{\kappa_4}{24} H_3(x)\phi(x) + \frac{\kappa_3^2}{72} H_5(x)\phi(x) \right\}}_{G(x)} + o(n^{-1}), \tag{11}$$

where $\kappa_3 = m_3/\sigma^3$ and $\kappa_4 = m_4/\sigma^4 - 3$ *is the skewness and kurtosis of $X_i$, respectively, $H_k(x)$ is the k-order Hermite polynomial, $\phi(x)$ is the density of $\Phi(x)$. The distribution $G(x)$ is called the EEs of $Z_n$. Furthermore,*

$$\sup_{x \in \mathbb{R}} |F_n(x) - G(x)| = o\left(n^{-1}\right).$$

The detailed proofs of Theorem 11 can be found in Section B of the Appendix. Intuitively, the Edgeworth expansions (11) have a similar form to the Taylor expansion, but are in a stochastic version. The distribution $G(x)$ from EEs is a more precise and absolutely consistent expansion of $F_n(x)$, including additional reminder terms of $O(n^{-1/2})$ and $O(n^{-1})$ on the basis of normal distribution $\Phi(x)$. For comparison, the remainder term for CLT in Theorem 3.1 is $o(1)$. The coefficients of the $n^{-1/2}$ and $n^{-1}$ term contain higher-order skewness and kurtosis information of relative scores, which more accurately characterizes the approximation error from $\Phi(x)$ to $F_n(x)$.

Theorem 4.1 enables a more accurate determination of the better generative model compared to the CLT version (7). Similarly, we can obtain the following $(1 - \alpha)$-level confidence interval $\widehat{\text{CI}}_{\text{EEs}}(\alpha)$:

$$\left[\hat{\delta}(\hat{\mathbb{P}}_1, \hat{\mathbb{P}}_2) - \alpha_1\sqrt{\frac{V}{n}}, \ \hat{\delta}(\hat{\mathbb{P}}_1, \hat{\mathbb{P}}_2) - \alpha_2\sqrt{\frac{V}{n}}\right], \tag{12}$$

where $\alpha_1, \alpha_2$ are two quantiles of $G(x)$ resulting in the shortest CI length of $\widehat{\text{CI}}_{\text{EEs}}$, satisfying $\int_{\alpha_1}^{\alpha_2} \mathrm{d}G(x) = 1 - \alpha$, s.t. $\mathrm{d}G(\alpha_1) = \mathrm{d}G(\alpha_2)$.

In practice, the variance of relative scores $\log(\hat{p}_1(Y_i)) - \log(\hat{p}_2(Y_i))$ is usually unknown, one can consider the $T$-statistics $T_n = (\hat{V}/n_{\mathcal{D}_{\text{test}}})^{-1/2}[\hat{\delta}(\hat{\mathbb{P}}_1, \hat{\mathbb{P}}_2) - \delta(\hat{\mathbb{P}}_1, \hat{\mathbb{P}}_2)]$, similar to Corollary 3.1. The details of EEs of $T_n$ and the counterpart bias-corrected CI are given in Section B of the Appendix[7]

## 5 Numerical analysis

### 5.1 Unconditional comparison

#### 5.1.1 Simulated data

We use $\mathbb{P}, \hat{\mathbb{P}}_1, \hat{\mathbb{P}}_2$ to denote the distribution of $Y, Y_1$, and $Y_2$ and generate the data by:

$$
\begin{aligned}
&X, X_1, X_2 \sim \mathcal{N}(0, I_d), \\
&Y \sim AX + B, \quad Y_1 \sim AX_1 + B, \\
&Y_2 \sim (A + \epsilon I_d)X_2 + B + \epsilon,
\end{aligned}
\tag{13}
$$

where $d = 10$, $A \in \mathbb{R}^{d \times d}$ is a constant diagonal matrix with diagonal elements generated from $\text{U}(0.8, 1.2)$, $B \in \mathbb{R}^d$ is a constant vector generated from $\mathcal{N}(0, I_d)$, and $\epsilon \in \mathbb{R}$ is a constant controls the difference between $\hat{\mathbb{P}}_1$ and $\hat{\mathbb{P}}_2$. We consider $\epsilon \in \{0.01, 0.02, \ldots, 0.2\}$, generate $n = 1000$ sample from $Y$, and construct the confidence interval via (7) with $\alpha = 0.1$[8]. In this setting, we deliberately design $Y, Y_1$, and $Y_2$ to be multivariate normal, since this is the only multivariate setting in which the Wasserstein-2 distance admits a closed-form expression, which is used by the coverage rate and power calculation below.

We compute the coverage rate and power of our confidence interval over 1000 repeated experiments. The results are shown in Figure 3. Our confidence intervals achieve coverage rates close to the target level. For comparison, we estimate $-\text{KL}(\mathbb{P}\|\hat{\mathbb{P}}_1)$ and $-\text{KL}(\mathbb{P}\|\hat{\mathbb{P}}_2)$ *separately* using the $k$ nearest neighbor (kNN) based estimator in Zhao & Lai (2020), take their difference, and construct confidence intervals for $-\text{KL}(\mathbb{P}\|\hat{\mathbb{P}}_1) + \text{KL}(\mathbb{P}\|\hat{\mathbb{P}}_2)$ using resampling methods including Subsampling (Politis & Romano, 1994) and Adaptive HulC (Kuchibhotla et al., 2024). We also examine the estimation and resampling-based inference

---

[7]When the sample size $n$ is relatively small, the EEs can provide more accurate statistical inference results, such as a lower coverage error for confidence intervals, lower type-I error for hypothesis testing, and so on. We leave these theoretical problems for further research.

[8]As discussed in Section 4.2, the sample size $n = 1000$ is relatively large, therefore we consider the CI using standard CLT.

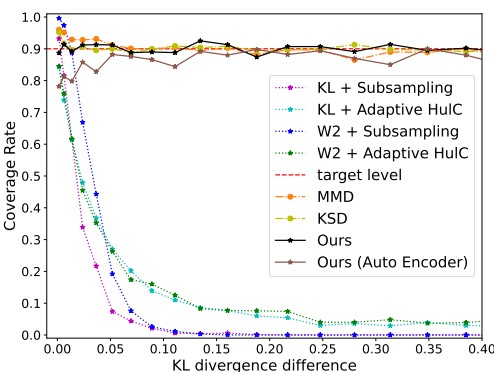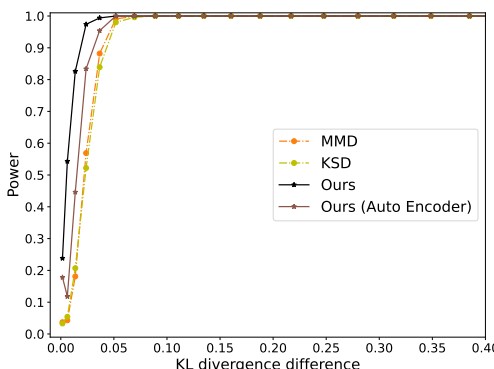

Figure 3: Coverage rate and power of confidence intervals constructed by our methods (7) and existing KL divergence and $W_2$ distance estimator paired with resampling methods (Subsampling and and Adaptive HulC). We provide two implementations of our procedure: "Ours" can access $\hat{p}_1$, $\hat{p}_2$, while "Ours (auto-encoder)" uses an auto-encoder to approximate $\hat{p}_1$, $\hat{p}_2$ (further details are provided in Section D of the Supplementary Material).

of the Wasserstein-2 distance difference, $-W_2^2(\mathbb{P}, \mathbb{P}_1) + W_2^2(\mathbb{P}, \mathbb{P}_2)$, where each Wasserstein-2 distance is estimated using the empirical distributions. As illustrated in Figure 3, these methods fail to provide faithful confidence intervals. More discussion on why existing estimators fail to provide valid confidence intervals can be found in Section D.2

In addition, we compare our procedure with other kernel-based methods, where the asymptotic distributions of the estimators are well established. Specifically, we consider the kernel Stein discrepancy (KSD) (Kanagawa et al., 2023) and maximum mean discrepancy (MMD) (Bounliphone et al., 2016). For KSD, we use the estimator in Kanagawa et al. (2023) and use the jackknife method to estimate the variance. For MMD, we use the estimator and variance estimator in Bounliphone et al. (2016). We compute the coverage rate and power of the confidence intervals over 1000 repeated experiments. As shown in Figure 3, the kernel-based methods provide valid confidence intervals, but their powers are lower compared to our methods.

In Section D of the Supplementary Materials, we illustrate the inference performance of our proposed CLT and EEs methods on more complicated data generation processes. Across various input distributions of $X$ and various smooth output models of $Y$, the two methods are similar and achieve desirable coverage for a moderately large test sample. For small test datasets, EEs correct the coverage rate to $1 - \alpha$ without sacrificing too much CI length and power, realizing reliability and detection ability trade-off of generative model comparison when the number of data is small. More discussion on when to use CLT intervals versus EEs intervals are provided in Section D.

### 5.1.2 Generative models for image

We apply our methods to real image datasets: evaluating generative models on the CIFAR-10 dataset. We consider denoising diffusion implicit model (DDIM) (Song et al., 2021), normalizing flow (NF) model Dinh et al. (2016), and variational auto encoder (VAE) model (Kingma, 2013). For DDIM, we consider the deterministic forward pass (see e.g. Section 4.3 of Song et al. (2021)) as the inverse transformation of the generative model. We compare pretrained models from Song et al. (2021) with different numbers of denoising steps, $S = 20, 50, 100$, and denote the corresponding generative distribution by $\hat{\mathbb{P}}_{\text{DDIM}_S}$. Following Song et al. (2021), we select the sub-sequence time steps using the quadratic schedule. For the NF and VAE models, we trained them using the default settings provided in the respective repositories[9].

Our results are consistent with those of Song et al. (2021) based on FID. Specifically, Song et al. (2021) shows that DDIM models with a larger number of denoising steps $S$ achieve better FID scores, our method also indicates that larger $S$ results in lower KL divergence. Moreover, while the FID scores of $\hat{\mathbb{P}}_{50}$ and $\hat{\mathbb{P}}_{100}$ are similar, our results indicate that DDIM model with $S = 100$ is significantly better.

---

[9]`https://github.com/chrischute/real-nvp`, `https://github.com/AntixK/PyTorch-VAE`

Table 1: Comparison of VAE model, NF model and DDIM model with different number of denoising steps $S$. For DDIM models, a larger number of denoising steps $S$ leads to a model with better FID scores and KL divergence. Our confidence interval doesn't cover 0, indicating $\hat{\mathbb{P}}_{100}$ is significantly better than both $S = 20$ and $S = 50$.

| Model | FID | $\widehat{\mathrm{CI}}$ of $\delta(\hat{\mathbb{P}}_M, \hat{\mathbb{P}}_{\mathrm{DDIM}_{100}})$ | $\widehat{\mathrm{CI}}$ of relative KID |
|---|---|---|---|
| VAE | 175.68 | (-8556.13, -8422.06) | (-1.6107e-01, -1.5412e-01) |
| NF | 83.26 | (-147888.77, -118430.73) | (-7.8398e-02, -7.2446e-02) |
| $\mathrm{DDIM}_{20}$ | 6.84 | (-39.91, -38.70) | (-2.5521e-03, -1.4817e-03) |
| $\mathrm{DDIM}_{50}$ | 4.67 | (-17.40, -16.63) | (-2.3001e-04, 1.5825e-04) |
| $\mathrm{DDIM}_{100}$ | 4.16 | - | |

Table 2: Comparison of GPT-2 model variants on the WikiText-2 dataset. Perplexity values are taken from the public leaderboard. Our results show that larger models achieve better performance in terms of KL divergence, consistent with the standard perplexity metric. The confidence interval for $\delta(\hat{\mathbb{P}}\mathrm{GPT}\ (\mathrm{FP16}), \hat{\mathbb{P}}\mathrm{GPT2})$ includes zero, indicating no significant difference between GPT-2 and its quantized version. This supports the practical use of the quantized model.

| Model | Perplexity | $\widehat{\mathrm{CI}}$ of $\delta(\hat{\mathbb{P}}_M, \hat{\mathbb{P}}_{\mathrm{GPT2}})$ |
|---|---|---|
| GPT2-Small | 29.41 | (67.763, 71.609) |
| GPT2-Medium | 22.76 | (28.874, 30.652) |
| GPT2-Large | 19.93 | (12.413, 13.351) |
| GPT2 (FP16) | – | (-0.002, 0.008) |
| GPT2 | 18.34 | – |

For comparison, we also evaluate the relative Kernel Inception Distance (Bińkowski et al., 2018), which evaluates the maximum mean discrepancy (MMD) with a third order polynomial kernel, computed on InceptionV3 (Szegedy et al., 2015) features of the images. We follow Bounliphone et al. (2016) to compute the relative KID scores and construct the confidence intervals. In the comparison between $\mathrm{DDIM}_{50}$ and $\mathrm{DDIM}_{100}$, the confidence interval for KID covers 0, which fails to distinguish these two models. These results are consistent with our observations in Section 5.1.1, that our method has better power compared to kernel-based method.

### 5.1.3 Generative models for text

We compare variants of the pre-trained GPT-2 model on the Wikitext-2 data. Particularly, we compare GPT-2 model with different sizes and quantization (models with 12, 24, 36, 48 layers are denoted as GPT2-small, GPT2-medium, GPT2-large and GPT2 respectively, and models with FP16 quantization are denoted by GPT2 (FP16)). We use the test set of the Wikitext-2 data, treating each non-title, non-empty line as a data point, estimate the relative KL divergence by (5), and construct the confidence intervals via (7). The results are summarized in Table 2.

Our results yield a model ranking that is consistent with the ordering based on the perplexity. But in contrast to perplexity, our framework further provides uncertainty quantification.

Notably, the performances of GPT2 (FP16) and GPT2 are statistically indistinguishable under our metric, indicating that the observed gap is not significant. Our result is consistent with the common practice of using quantization to improve efficiency without sacrificing performance.

### 5.2 Conditional comparison

We compare three LLMs, OPT-1.3B (Zhang et al., 2022), Mistral-7B (Jiang et al., 2023), and Llama3-8B (Dubey et al., 2024) on the validation data of the TriviaQA data set. We treat each language model as a conditional generative model that models the conditional distribution $p(\mathrm{Answer}|\mathrm{Question})$. The detailed settings are given in Section D of the Appendix. The results are summarized in Table 3. Our results are consistent with commonly used metrics such as the F1 score.

We further investigate how sample size impacts the confidence intervals of our method by constructing confidence intervals using a subset of the validation data. Figure 4 shows the comparison between Mistral-

Table 3: Comparison of language models on the TriviaQA dataset.

| Model | F1 score | $\widehat{\text{CI}}$ of $\delta(\hat{\mathbb{P}}_M, \hat{\mathbb{P}}_{\text{Llama3}-8\text{B}})$ |
|---|---|---|
| OPT-1.3B | 7.47 | (-7.19, -7.01) |
| Mistral-7B | 12.24 | (-0.22, -0.14) |
| Llama3-8B | 36.34 | – |

7B and Llama3-8B. Even with a relatively small sample size (e.g., $n = 100$), our method (both CLT and EEs-based) can identify the better model with statistical significance. Since the comparison starts with a very small test sample, our EEs procedure is more appropriate in the starting stage (e.g., for $n = 50$, the CLT-based interval includes zero, whereas the EEs-based interval does not).

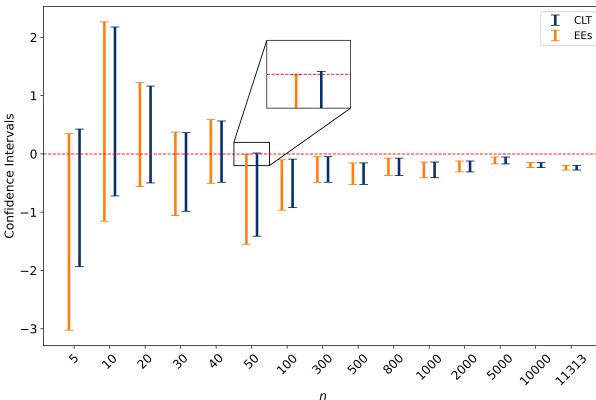

Figure 4: Confidence intervals of Llama3-8B vs. Mistral-7B for various sample size $n$

## 6 Discussions

In this paper, we proposed a model-free and nuisance-free approach for quantitatively comparing generative models with statistical confidence. First, we propose focusing on the relative performance gap (relative score) between two generative models, rather than evaluating their absolute performances individually. Second, we developed an unbiased estimator for the relative score that achieves parametric convergence rates and is asymptotically normal, enabling rigorous inference. Third, on simulated datasets with known ground truth, our method effectively controls type I error while achieving comparable or superior power, whereas existing metrics often exhibit near-zero coverage; when applied to generative models on real image or text datasets, our approach yields statistically confident conclusions consistent with existing metrics but with uncertainty quantification.

The statistical significance under the proposed relative-KL metric should not be over-interpreted. The comparison only concerns whether the distribution of one generative model is closer to the true data distribution than the distribution of another generative model in terms of the KL-divergence. It doesn't directly imply other properties such as safety. When safety and harmful behaviors are the main concerns, specific metrics should be designed, and our method can serve as a complementary metric.

## Acknowledgement

Portions of this work was performed on High Performance Computing resources provided by the Advanced Research Computing Services team at NJIT. Han Su was supported by the Doctoral Student Program of the Young S&T Talents Cultivation Project, CAST.

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

## Appendix

**Overview**. The appendix is organized as follows. Section A reviews related literature on distance measures between distributions and quantitative evaluation metrics for image generative models. Section B provides theoretical proofs of the main results. Section C outlines computational and implementation details. Section D presents additional empirical results and further descriptions of the simulation setups introduced in the main text.

## A   Literature

### A.1   Dissimilarity metrics between distributions

Quantitative evaluation of generative models typically involves computing some distance between probability distributions. In this section, we review commonly used dissimilarity metrics between distributions and evaluation metrics of generative models.

#### A.1.1   f-Divergence

For a convex function $f : [0, +\infty) \to (-\infty, +\infty]$ such that $f(x)$ is finite for all $x > 0$, $f(1) = 0$, and $f(0) = \lim_{t \to 0^+} f(t)$, the f-divergence of $P$ from $Q$ is given by

$$D_f(\mathbb{P} \| \hat{\mathbb{P}}_1) \equiv \int_\Omega f\left( \frac{p(y)}{\hat{p}_1(y)} \right) \hat{p}_1(y) dy$$

KL-divergence is a special case of f-divergence with $f(x) = x \log(x)$.

Estimating $f$-divergence typically requires the estimation of two densities, $p$ and $\hat{p}_1$, where the estimation of $p$ exposes us to the curse of dimensionality. For the associated relative score, the convenient cancellation that simplifies computation is specific to the KL divergence (Proposition 3.1). Consequently, for divergences other than KL, estimating the density $p$ becomes unavoidable, and the estimation of both absolute and relative $f$-divergences no longer achieves the parametric rate.

#### A.1.2   Wasserstein-p Distance

Wasserstein-$p$ distance (Villani et al., 2009) is defined as

$$W_p(\mathbb{P}, \hat{\mathbb{P}}_1) = \inf_{\gamma \in \Gamma(\mathbb{P}, \hat{\mathbb{P}}_1)} \mathbb{E}^{1/p}_{(x,y) \sim \gamma} \left[ \| x - y \|_p^p \right],$$

where $\Gamma(\mathbb{P}, \hat{\mathbb{P}}_1)$ is the set of couplings between $\mathbb{P}$ and $\hat{\mathbb{P}}_1$.

We next discuss the challenges of performing inference for evaluation methods based on the Wasserstein-$p$ distance. First, Del Barrio & Loubes (2019) shows that[10]

$$\sqrt{n} \left( W_2^2(\mathbb{P}_n, \hat{\mathbb{P}}_{1,n}) - \mathbb{E}[W_2^2(\mathbb{P}_n, \hat{\mathbb{P}}_{1,n})] \right) \xrightarrow{d} \mathcal{N}\left( 0, \sigma^2(\mathbb{P}, \hat{\mathbb{P}}_1) \right),$$

where the asymptotic variance $\sigma^2(\mathbb{P}, \hat{\mathbb{P}}_1)$ can be estimated consistently using a plug-in estimator. The issue is that the center $\mathbb{E}[W_2^2(\mathbb{P}_n, \hat{\mathbb{P}}_{1,n})]$ is different from the desired $W_2^2(\mathbb{P}, \hat{\mathbb{P}}_1)$, and the gap between $\mathbb{E}[W_2^2(\mathbb{P}_n, \hat{\mathbb{P}}_{1,n})]$ and $W_2^2(\mathbb{P}, \hat{\mathbb{P}}_1)$ scales as $n^{-1/d}$ (Villani et al., 2009). Second, for the relative Wasserstein-2 distance $W_2^2(\mathbb{P}, \hat{\mathbb{P}}_1)$, the joint asymptotic distribution of $(W_2^2(\mathbb{P}_n, \hat{\mathbb{P}}_{1,n}), W_2^2(\mathbb{P}_n, \hat{\mathbb{P}}_{2,n}))$ is required. It remains unclear whether $W_2^2(\mathbb{P}_n, \hat{\mathbb{P}}_{1,n})$ and $W_2^2(\mathbb{P}_n, \hat{\mathbb{P}}_{2,n})$ are asymptotically jointly Gaussian, as well as what the exact form of their covariance matrix is[11]. Third, subsampling methods (Dümbgen, 1993) can be employed for conducting inference for Wasserstein-p distances; however, subsampling is computationally expensive, especially given that the Wasserstein-p distance is already difficult to compute. For a comprehensive review of these issues, see Panaretos & Zemel (2019).

---

[10]CLT results of general Wasserstein-$p$ distance are largely unknown.

[11]The off-diagonal values of the covariance matrix is non-zero because both $W_2^2(\mathbb{P}_n, \hat{\mathbb{P}}_{1,n})$ and $W_2^2(\mathbb{P}_n, \hat{\mathbb{P}}_{2,n})$ depend on $\mathbb{P}_n$.

### A.2 Existing evaluation metrics of generative models for image

We describe the inception score (IS) (Salimans et al., 2016) and the Frechet Inception Distance (FID) (Heusel et al., 2017), two most commonly used quantitative scores for generative models. We note that these metrics were originally designed for training GANs rather than for model evaluation, with more emphasis placed on computational efficiency than on statistical rigor.

IS evaluates the quality of a generative model by applying a separate, pretrained image classification model to a batch of images generated by the model. IS is maximized when the classifier confidently predicts a single label for each image, or when the predictions are evenly distributed across all possible labels. The quality of IS depends heavily on the quality of the classifier (if the classifier consistently outputs a single label for all images, the IS becomes uninformative). Another disadvantage is that IS does not compare generated images to test images.

FID compares the distribution between the distribution of test images and that of generated images. Mathematically, FID is defined as the Wasserstein-2 distance between the two distributions. However, the Wasserstein-2 distance is computationally expensive for random vectors, except for multivariate Gaussians. To approximate the FID, the default approach involves two steps: (1) mapping the real and generated images to $\mathbb{R}^d$ separately by passing them through the final layer of an image classifier to extract essential features; (2) fitting multivariate Gaussian distributions to the transformed data in $\mathbb{R}^d$ and computing the Wasserstein-2 distance between these multivariate Gaussians. The approximation accuracy depends on how well the transformation to $\mathbb{R}^d$ captures the data characteristics and the quality of the multivariate Gaussians fit to the transformed data (the covariance matrix is typically not diagonal and the estimation involves $O(d^2)$ elements). Chong & Forsyth (2020) shows that FID could be significantly biased in finite sample.

## B Proof

### B.1 Proof of results in Section 3

*Proof of Proposition 1.* For simplicity, we use $\mathbb{P}_1$ instead of $\hat{\mathbb{P}}_1$, $p_1$ instead of $\hat{p}_1$, and similarly for $\mathbb{P}_2$ and $p_2$. For any densities $p(x)$, $p_1(x)$, $p_2(x)$, we define

$$h(p, p_1, p_2) := \int f\left(\frac{p_1(x)}{p(x)}\right) - f\left(\frac{p_2(x)}{p(x)}\right) p(x) dx.$$

For any $\delta(x)$ such that $\int \delta(x) dx = 0$, and $t$ such that $p(x) + t\delta(x) \geq 0$, by the calculus of variations and (4),

$$
\begin{aligned}
\int g(p_1(x), p_2(x)) \delta(x) dx &= \frac{\partial}{\partial t} h(p + t\delta, p_1, p_2)|_{t=0} \\
&= \int \left(-f'\left(\frac{p_1(x)}{p(x)}\right)\frac{p_1(x)}{p^2(x)} + f'\left(\frac{p_2(x)}{p(x)}\right)\frac{p_2(x)}{p^2(x)}\right)\delta(x)p(x)dx \\
&\quad + \int \left(f\left(\frac{p_1(x)}{p(x)}\right) - f\left(\frac{p_2(x)}{p(x)}\right)\right)\delta(x)dx \\
&= \int -f'\left(\frac{p_1(x)}{p(x)}\right)\frac{p_1(x)}{p(x)}\delta(x) + f'\left(\frac{p_2(x)}{p(x)}\right)\frac{p_2(x)}{p(x)}\delta(x) \\
&\quad + f\left(\frac{p_1(x)}{p(x)}\right)\delta(x) - f\left(\frac{p_2(x)}{p(x)}\right)\delta(x)dx.
\end{aligned}
\tag{14}
$$

We let $f_1(x) = f(e^x)$, then $f_1(\log(x)) = f(x)$ and $f_1'(\log(x)) \cdot (1/x) = f'(x)$, which implies $f_1'(\log(x)) = xf'(x)$. We replace $f(x)$ by $f_1(x)$ in Eq. (14),

$$
\begin{aligned}
\int g(p_1(x), p_2(x)) \delta(x) dx = \int &\left(-f_1'\left(\log\left(\frac{p_1(x)}{p(x)}\right)\right) + f_1\left(\log\left(\frac{p_1(x)}{p(x)}\right)\right)\right. \\
&\left. + f_1'\left(\log\left(\frac{p_2(x)}{p(x)}\right)\right) - f_1\left(\log\left(\frac{p_2(x)}{p(x)}\right)\right)\right)\delta(x)dx.
\end{aligned}
\tag{15}
$$

Since Equation (15) is true for arbitrary $\delta(x)$ such that $\int \delta(x) = 0$, then we let $f_2(x) = -f_1'(x) + f_1(x)$ and have

$$g(p_1(x), p_2(x)) = f_2\left(\log(p_1(x)) - \log(p(x))\right) - f_2\left(\log(p_2(x)) - \log(p(x))\right) + C, \quad \forall x. \tag{16}$$

for some constant $C \in \mathbb{R}$. We assume $C = 0$, otherwise, we replace $g(p_1(x), p_2(x))$ by $g(p_1(x), p_2(x)) - C$. Note that $f_2(0) = -f_1'(0) + f_1(0) = -f_1'(0) + f(1) = -f_1'(0)$. We let $f_3(x) = f_2(x) + f_1'(0)$, then $f_3(0) = 0$ and $g(p_1(x), p_2(x)) = f_3\left(\log(p_1(x)) - \log(p(x))\right) - f_3\left(\log(p_2(x)) - \log(p(x))\right)$. Take $p(x) = p_1(x)$ in Eq. (16),

$$g(p_1(x), p_2(x)) = f_3(0) - f_3\left(\log(p_2(x)) - \log(p_1(x))\right) = f_3\left(\log(p_2(x)) - \log(p_1(x))\right). \tag{17}$$

We let $a = \log(p_2(x)) - \log(p_1(x))$, $b = \log(p(x)) - \log(p_2(x))$, then by (17),

$$f_3(a + b) - f_3(b) = f_3(a). \tag{18}$$

Since $f \in C^1$, then $f_3$ is continuous. By the result of Cauchy's functional equation, there exists $c, d \in \mathbb{R}$ such that

$$f_3(x) = cx + d, \quad \forall x. \tag{19}$$

By the definition of $f_3(x)$ and (19), we have $f_2(x) = f_3(x) - f_1'(0) = cx + d'$ for some $d' \in \mathbb{R}$. By the definition of $f_2(x)$,

$$-f_1'(x) + f_1(x) = cx + d'.$$

This is a standard ODE problem, and we multiply both sides by $e^{-x}$ to get

$$
\begin{aligned}
&- \left(e^{-x} f_1(x)\right)' = cxe^{-x} + d'e^{-x} \\
&\implies e^{-x} f_1(x) = -cxe^{-x} - ce^{-x} - d'e^{-x} + d'' \\
&\implies f_1(x) = \alpha e^x + \beta x + \theta,
\end{aligned}
$$

where $d'', \alpha, \beta, \theta \in \mathbb{R}$. By the definition of $f_1(x)$, $f(x) = \alpha x + \beta \log(x) + \theta$. Note that $f(1) = 0$, which implies $\alpha + \theta = 0$. Note that $\int (\alpha(d\mathbb{P}_1/d\mathbb{P}) - \alpha)d\mathbb{P} = 0$, therefore, the divergence with $f(x) = \alpha x + \beta \log(x) + \theta$ is the same as that of $\beta \log(x)$. Since $f(x)$ is convex, we have $\beta \leq 0$, and we finish the proof. $\qquad\square$

## B.2 Proof of results in Section 3.1

*Proof of Proposition 2.* Proposition 2 follows from the linearity of expectation and the fact that the test data $Y_i \sim \mathbb{P}$. $\qquad\square$

*Proof of Theorem 3.1.* Recall that $Y_i$ are i.i.d. sampled from $\mathbb{P}$. Theorem 3.1 is an application of the Lindeberg-Lévy central limit theorem to i.i.d. random variables $\log(\hat{p}_1(Y_i)) - \log(\hat{p}_2(Y_i))$. $\qquad\square$

*Proof of Corollary 3.1.* When $0 < V < \infty$, by the law of large numbers of i.i.d. random variables, the empirical variance $\hat{V}$ of $\log(\hat{p}_1(Y_i)) - \log(\hat{p}_2(Y_i))$ converges to $V$ in probability. We further combine Slutsky's theorem and Corollary 3.1 to arrive at Theorem 3.1. $\qquad\square$

*Proof of Corollary 3.2.* Corollary 3.2 comes from the definition of convergence in distribution and Corollary 3.1. $\qquad\square$

## B.3 Proof of results in Section 4.2 and additional discussions on EEs

The following assumptions ensure the existence of EEs of $Z$-statistics

**Assumption B.1.** *Suppose the relative scores $X_i$ have high-order moments, $\mathbb{E}|X|^4 < \infty$.*

**Assumption B.2.** *Suppose that the distribution $F$ of relative scores admits absolutely continuous components, that is, $\limsup_{|t| \to \infty} |\psi_X(t)| < 1$.*

*Proof of Theorem 4.1.* Under Assumptions B.1 and B.2, the EEs of $Z_n$ can be conducted in three steps: (1) Calculation of cumulants; (2) High-order expansion of c.f. $\psi_{Z_n}(t)$; (3) The Fourier inversion of c.f. $\psi_{Z_n}(t)$.

**Step 1: The definition and calculation of cumulants.** The definition of the cumulants of a random variable $X$ is related to the polynomial expansion of its logarithmic characteristic function. Specifically, we first consider the logarithmic transformation of $\psi_X(t)$, i.e.,

$$\log \psi_X(t) = \log \left\{ 1 + \sum_{l \geq 1} \frac{1}{l!} \mu_l (it)^l \right\} := \sum_{l \geq 1} \frac{1}{l!} c_l (it)^l,$$

where $\{\mu_l\}_{l \geq 1}$ are the raw moments of $X$, and $\{c_l\}_{l \geq 1}$ are defined as the *cumulants* of $X$, obtained by expanding the $\log(1 + a)$ terms in the above equation via the Taylor series and matching the corresponding coefficients of polynomials. By simple calculations, the first four cumulants of $X$ are defined as:

$$c_1 = \mu, \qquad c_2 = \mu_2 - \mu^2 = \sigma^2,$$
$$c_3 = \mu_3 - 3\mu_2\mu + 2\mu^3 = E(X - \mu)^3 := m_3,$$
$$c_4 = \mu_4 - 4\mu_3\mu - 3\mu_2^2 + 12\mu^2\mu_2 - 6\mu^4 = E(X - \mu)^4 - 3\sigma^4 := m_4 - 3\sigma^4$$

where $m_3$ and $m_4$ are the third and fourth central moments of $X$, respectively, indicating the close relationship between cumulants and moments. Thus, $\psi_X(t)$ can be expressed as an exponential function of the cumulants $\{c_l\}_{l \geq 1}$, shown as follows:

$$\psi_X(t) = \exp\{\log \psi_X(t)\} = \exp\left\{ itc_1 + \frac{(it)^2}{2} c_2 + \frac{(it)^3}{3!} c_3 + \frac{(it)^4}{4!} c_4 + o\left(t^4\right) \right\}.$$

**Step 2: Expansion of the Characteristic Function.** Similarly, the logarithmic characteristic function $\psi_Y(t)$ of the standardized variable $Y = (X - \mu)/\sigma$ can also be expressed in terms of the cumulants $\{\kappa_l\}_{l \geq 1}$ of $Y$:

$$\psi_Y(t) = \exp\left\{ it\kappa_1 + \frac{(it)^2}{2} \kappa_2 + \frac{(it)^3}{3!} \kappa_3 + \frac{(it)^4}{4!} \kappa_4 + o\left(t^4\right) \right\}, \tag{20}$$

where $\kappa_1 = 0$, $\kappa_2 = 1$, $\kappa_3 = m_3/\sigma^3$ is the skewness of $X$, and $\kappa_4 = m_4/\sigma^4 - 3$ is the kurtosis of $X$.

**Step 3: Inversion of the Distribution Function.** According to (20), the characteristic function $\psi_{Z_n}(t)$ of $Z_n$ in its expansion of CLT version (9) can be rewritten as a function of the cumulants as follows:

$$\begin{aligned}
\psi_{Z_n}(t) &= \psi_Y^n \left( \frac{t}{\sqrt{n}} \right) \\
&= \exp\left\{ n \left( -\frac{1}{2} \frac{t^2}{n} + \frac{1}{3!} \kappa_3 \left( \frac{it}{\sqrt{n}} \right)^3 + \frac{1}{4!} \kappa_4 \left( \frac{it}{\sqrt{n}} \right)^4 + o\left( \frac{1}{n^2} \right) \right) \right\} \\
&= \exp\left\{ -\frac{t^2}{2} + \frac{1}{3!} \kappa_3 \frac{(it)^3}{\sqrt{n}} + \frac{1}{4!} \kappa_4 \frac{(it)^4}{n} + o\left( \frac{1}{n} \right) \right\} \\
&= e^{-t^2/2} \exp\left\{ \frac{1}{3!} \kappa_3 \frac{(it)^3}{\sqrt{n}} + \frac{1}{4!} \kappa_4 \frac{(it)^4}{n} + o\left( \frac{1}{n} \right) \right\} \\
&= e^{-t^2/2} \left\{ 1 + \frac{1}{3!} \kappa_3 \frac{(it)^3}{\sqrt{n}} + \frac{1}{4!} \kappa_4 \frac{(it)^4}{n} + \frac{1}{2 \times (3!)^2} \kappa_3^2 \frac{(it)^6}{n} + o\left( \frac{1}{n} \right) \right\}. \tag{21}
\end{aligned}$$

Furthermore, by inverting the distribution function via Fourier inversion, we can obtain the Edgeworth expansion results. Specifically, let $\Phi^{(j)}(x)$ denote the $j$th derivative of $\Phi(x)$. Noting that the characteristic function is the Fourier transformation of the distribution function, along with the formula of integration by

parts and some straightforward calculations, we obtain the following series of results:

$$
\begin{aligned}
e^{-t^2/2} &= \int_{-\infty}^{\infty} e^{itx} \mathrm{d}\Phi(x) \\
&= \int_{-\infty}^{\infty} \frac{1}{it} \Phi^{(1)}(x) \mathrm{d}e^{itx} \\
&= \frac{1}{it} \left\{ \Phi^{(1)}(x)e^{itx} \Big|_{-\infty}^{\infty} - \int_{-\infty}^{\infty} e^{itx} \mathrm{d}\Phi^{(1)}(x) \right\} \\
&= (-it)^{-1} \int_{-\infty}^{\infty} e^{itx} \mathrm{d}\Phi^{(1)}(x) \\
&= \cdots \\
&= (-it)^{-2} \int_{-\infty}^{\infty} e^{itx} \mathrm{d}\Phi^{(2)}(x) \\
&= \cdots \\
&= (-it)^{-j} \int_{-\infty}^{\infty} e^{itx} \mathrm{d}\Phi^{(j)}(x),
\end{aligned}
$$

which indicates that $(-it)^j e^{-t^2/2}$ is the Fourier transform of $\Phi^{(j)}(x)$. Denoting the differential operator as $D$, we have

$$
\int_{-\infty}^{\infty} e^{itx} \mathrm{d}(-D)^j \Phi(x) = (it)^j e^{-t^2/2}. \tag{22}
$$

Applying the differential operator repeatedly to $\Phi(x)$, we can obtain that

$$
\begin{aligned}
D\Phi(x) &= \phi(x) := H_0(x)\phi(x), \\
D^2\Phi(x) &= -x\phi(x) := -H_1(x)\phi(x), \\
D^3\Phi(x) &= (x^2 - 1)\phi(x) := H_2(x)\phi(x), \\
D^4\Phi(x) &= -(x^3 - 3x)\phi(x) := -H_3(x)\phi(x), \\
&\vdots
\end{aligned}
$$

In sum up, we have

$$
(-D)^j \Phi(x) = -H_{j-1}(x)\phi(x), \quad j = 0, 1, 2, \cdots \tag{23}
$$

where $\{H_j(x)\}_{j \geq 0}$ are the Hermite polynomials, satisfying the following properties: (1) $\{H_j(x)\}_{j \geq 0}$ are orthogonal with respect to $\phi(x)$, i.e., $\int_{-\infty}^{\infty} H_j(x)H_k(x)\phi(x)\mathrm{d}x = 0, j \neq k$; (2) $H_j(x)$ is a polynomial of degree $j$, and its parity (even/odd) matches the parity of the index $j$.

Naturally, according to (21)–(23), the characteristic function $\psi_{Z_n}(t)$ of $Z_n$ can be viewed as the Fourier transform of the following distribution:

$$
\begin{aligned}
&\Phi(x) + \frac{\kappa_3}{6\sqrt{n}}(-D)^3\Phi(x) + \frac{\kappa_4}{24n}(-D)^4\Phi(x) + \frac{\kappa_3^2}{72n}(-D)^6\Phi(x) + o\left(\frac{1}{n}\right) \\
=&\Phi(x) - n^{-1/2}\frac{\kappa_3}{6}H_2(x)\phi(x) - n^{-1}\left\{\frac{\kappa_4}{24}H_3(x)\phi(x) + \frac{\kappa_3^2}{72}H_5(x)\phi(x)\right\} + o\left(n^{-1}\right). \tag{24}
\end{aligned}
$$

This indicates that equation (24) represents the form of the distribution function of $Z_n$. Combining with the discussion in Section 4.2, equation (10) can be further written as

$$
F_n(x) = \Phi(x) - n^{-1/2}\frac{\kappa_3}{6}H_2(x)\phi(x) - n^{-1}\left\{\frac{\kappa_4}{24}H_3(x)\phi(x) + \frac{\kappa_3^2}{72}H_5(x)\phi(x)\right\} + o\left(n^{-1}\right). \tag{25}
$$

(25) is called the Edgeworth expansion for the $Z$-statistic. Furthermore, the result in (25) holds uniformly for $x \in \mathbb{R}$, that is,

$$\sup_{x \in \mathbb{R}} \left| F_n(x) - \left\{ \Phi(x) - n^{-1/2} \frac{\kappa_3}{6} H_2(x)\phi(x) - n^{-1} \left\{ \frac{\kappa_4}{24} H_3(x)\phi(x) + \frac{\kappa_3^2}{72} H_5(x)\phi(x) \right\} \right\} \right| = o\left(n^{-1}\right),$$

which can be similarly derived following Theorem 2.1 in Hall (2013), and we omit the details here. Thus, the proof of Theorem 4.1 is complete. $\qquad\square$

As the other side of coins, the variance of relative scores $\log(\hat{p}_1(Y_i)) - \log(\hat{p}_2(Y_i))$ is usually unknown in practice, one can consider the $T$-statistics $T_n = (\hat{V}/n_{\text{test}})^{-1/2}(\hat{\delta}(\hat{\mathbb{P}}_1, \hat{\mathbb{P}}_2) - \delta(\hat{\mathbb{P}}_1, \hat{\mathbb{P}}_2))$, similar to Corollary 3.1. Formally, the EEs of $T_n$ and its counterpart bias-corrected CI can be derived, as shown in the following theorem.

**Theorem B.1.** *The distribution function of $T_n$ satisfies*

$$F_n^*(x) = \Phi(x) + n^{-1/2}\frac{\kappa_3}{6}(2x^2+1)\phi(x) + \underbrace{n^{-1}\left\{ \frac{1}{12}\kappa_4 x(x^2-3) - \frac{1}{18}\kappa_3^2 x(x^4+2x^2-3) - \frac{1}{4}x(x^2+3) \right\}\phi(x)}_{G^*(x)} + o(n^{-1}).$$

(26)

*Furthermore, we can also derive the consistent convergence result of $T_n$*

$$\sup_{x \in \mathbb{R}} \left| F_n^*(x) - \left\{ \Phi(x) + n^{-1/2}\frac{\kappa_3}{6}(2x^2+1)\phi(x) + n^{-1}\left\{ \frac{1}{12}\kappa_4 x(x^2-3) - \frac{1}{18}\kappa_3^2 x(x^4+2x^2-3) - \frac{1}{4}x(x^2+3) \right\}\phi(x) \right\} \right|$$
$$= o\left(n^{-1}\right).$$

*Proof of Theorem B.1.* Similar to the proof of Theorem 4.1, we can obtain the high-order expansion term of $F_n^*(x)$ based on the calculation of cumulants, the expansion and Fourier inversion of c.f. of $T$-statistics. The detailed proofs can be found in Section 2.6 in Hall (2013), and we omit the details here. $\qquad\square$

Theorem B.1 also determins a $(1-\alpha)$-level confidence interval of the relative score based on $T$-statistics

$$\widehat{\text{CI}}_{\text{EEs}}^*(\alpha) := \left[ \hat{\delta}(\hat{\mathbb{P}}_1, \hat{\mathbb{P}}_2) - \beta_1 \sqrt{\frac{\hat{V}}{n}}, \ \hat{\delta}(\hat{\mathbb{P}}_1, \hat{\mathbb{P}}_2) - \beta_2 \sqrt{\frac{\hat{V}}{n}} \right], \tag{27}$$

where $\beta_1, \beta_2$ satisfies

$$\int_{\beta_1}^{\beta_2} \mathrm{d}G^*(x) = 1 - \alpha, \qquad \text{s.t. } \mathrm{d}G^*(\beta_1) = \mathrm{d}G^*(\beta_2).$$

Both (12) and (27) derived by the limiting distributions of Theorems 4.1 and B.1 are valid CIs, shown in the following statement.

**Corollary B.1.** *Under the Assumptions B.1 and B.2, for any $\alpha \in (0,1)$,*

$$\mathbb{P}\left(\delta(\hat{\mathbb{P}}_1, \hat{\mathbb{P}}_2) \in \widehat{\text{CI}}_{\text{EEs}}(\alpha)\right) = \mathbb{P}\left(\delta(\hat{\mathbb{P}}_1, \hat{\mathbb{P}}_2) \in \widehat{\text{CI}}_{\text{EEs}}^*(\alpha)\right) \to 1 - \alpha.$$

## C  Computation

Computing our relative score estimator involves the evaluation of the probability densities $\hat{p}_1(\cdot)$ and $\hat{p}_2(\cdot)$ at each test point. We provide explicit details on how this computation is conducted for generative models for image and text, respectively.

### C.1 Generative models for text

For models such as auto-regressive language models using transformers, $\hat{p}_1(\cdot)$ can be computed directly through a forward pass. The computational cost of our method is comparable to that of standard metrics such as perplexity, which also requires evaluations of the log-likelihood.

### C.2 Generative models for image

For generative models for images, samples are usually generated via an invertible transformation $g_1(\cdot)$[12] of a standard Gaussian variable $\mathcal{N}(0, I_{d_1})$. The density can be computed by mapping data points back to the latent space, where the latent embeddings follow a *known* multivariate normal distribution. Explicitly, let $g_1^{-1}$ be the inverse of $g_1$, then by the change of variable formula,

$$\log(\hat{p}_1(y)) = -\|g_1^{-1}(y)\|_2^2/2 - d_1 \log(\sqrt{2\pi}) + \log\left(|J_{g_1}|(g_1^{-1}(y))\right). \tag{28}$$

Similarly, we obtain (28) for $\log(\hat{p}_2(y))$. Then the estimator $\hat{\delta}(\hat{\mathbb{P}}_1, \hat{\mathbb{P}}_2)$ in (5) takes the form

$$\frac{1}{n_{\text{test}}} \sum_{i=1}^{n_{\text{test}}} \frac{1}{2}\left(\|g_2^{-1}(Y_i)\|_2^2 - \|g_1^{-1}(Y_i)\|_2^2\right) + (d_2 - d_1)\log(\sqrt{2\pi})$$
$$+ \log\left(|J_{g_1}|(g_1^{-1}(Y_i))\right) - \log\left(|J_{g_2}|(g_2^{-1}(Y_i))\right). \tag{29}$$

What remains is to determine the inverse functions, $g_1^{-1}$ and $g_2^{-1}$, as well as the corresponding Jacobian determinants, $|J_{g_1}|(g_1^{-1}(Y_i))$ and $|J_{g_2}|(g_2^{-1}(Y_i))$. Below, we discuss this computation over various generative models for images.

- For normalizing flows (Rezende & Mohamed, 2015; Dinh et al., 2016), the models are constructed to allow both the inverse transformation and the log-determinant of the Jacobian to be computed in closed form and evaluated efficiently.

- In auto-encoders, the encoder effectively serves the role of inverse of the decoder, i.e., $g^{-1}$.

- For diffusion models such as DDIM (Song et al., 2021), the forward process admits an inversion by solving the associated reverse-time ODE. Details in Section C.2.1.

**Remark 1** (Computation of the Jacobian determinant)**.** In our experiments, we mainly consider image datasets with low resolutions. For this setting, we compute the exact Jacobian log determinants using TORCH.AUTOGRAD.FUNCTIONAL.JACOBIAN. In settings with very high image resolution or an exceptionally large number of test data points, computation of the exact Jacobian log determinants may be infeasible. We suggest approximating the log-determinant of the Jacobian using Stochastic Lanczos Quadrature (SLQ) (Ubaru et al., 2017). Using the identity

$$\log|\det(J)| = \frac{1}{2}\log(J^\top J) = \frac{1}{2}\text{trace}(\log(J^\top J)),$$

the trace term can be estimated via stochastic trace estimation (Hutchinson, 1989):

$$\text{trace}(\log(J^\top J)) \approx \sum_{i=1}^{m} z_i^\top \log(J^\top J)z_i,$$

with random probes $z_i \sim \mathcal{N}(0, I)$. Each quadratic form $z_i^\top \log(J^\top J)z_i$ can be computed using Lanczos iterations (Ubaru et al., 2017), which only require matrix–vector products of the form $J^\top J z_i$. These products can be implemented using PyTorch automatic differentiation (e.g., TORCH.FUNC.JVP and TORCH.FUNC.VJP), with computational cost comparable to standard backpropagation. With SLQ, the overall cost of evaluating (28)

---

[12]For simplicity, we consider generative models with an encoder-decoder structure and treat the encoder as the inverse of the transformation $g_1(\cdot)$. As shown in Figure 3, despite using approximate likelihoods, the resulting confidence intervals achieve coverage rates close to the nominal level. We provide further sensitivity analysis in Section D.3.

is on the same order as training the model for one epoch on the evaluation dataset. We further empirically assess the approximation error of SLQ. Using the first test image of CIFAR-10 and the pretrained DDIM model in Section 5.1.2, we compute $\log\left(\left|\det(J_{g_1^{-1}})\right|\right)$ exactly using TORCH.AUTOGRAD.FUNCTIONAL.JACOBIAN, obtaining a value of 16177.51. We then apply SLQ with a single random probe and 10 Lanczos steps. Across 50 runs, SLQ yields a mean estimate of approximately 16189.15 with a standard deviation of 57.13. This demonstrates that SLQ provides a reasonably accurate approximation, even with a very small number of probes and iterations.

**Remark 2** (Equivalent forms of Jacobian determinant)**.** Since the Jacobian of the inverse function is the inverse matrix of the Jacobian of the original function, we have multiple equivalent forms for the term $\log(|J_{g_1}|(g_1^{-1}(Y_i))) - \log(|J_{g_2}|(g_2^{-1}(Y_i)))$ in (29),

$$\log\left(|J_{g_1}|(g_1^{-1}(Y_i))\right) - \log\left(|J_{g_2}|(g_2^{-1}(Y_i))\right)$$
$$= -\log\left(|J_{g_1^{-1}}|(Y_i)\right) + \log\left(|J_{g_2^{-1}}|(Y_i)\right)$$
$$= \log\left(|J_{g_1^{-1}\circ g_2}|(g_2^{-1}(Y_i))\right)$$
$$= \log\left(|J_{g_2^{-1}\circ g_1}|(g_1^{-1}(Y_i))\right).$$

Explicitly, $\log\left(|J_{g_1^{-1}}|(Y_i)\right)$ computes the log determinant of the Jacobian matrix of the inverse function $g_1^{-1}$, evaluated at the test data point $Y_i$; $\log\left(|J_{g_1^{-1}\circ g_2}|(g_2^{-1}(Y_i))\right)$ computes the log determinant of the composite function $g_1^{-1}\circ g_2$, evaluated at the latent embedding $g_2^{-1}(Y_i)$ of the test data point under the second generative model. In practice, either of these equivalent formulations can be chosen based on which Jacobian determinants can be computed more efficiently and accurately.

In practice, the choice of form depends on whose Jacobian determinants can be computed more efficiently and accurately: the original $(g_1, g_2)$, the inverse $(g_1^{-1}, g_2^{-1})$, or the composite $(g_1^{-1}\circ g_2)$.

### C.2.1 DDIM

The log of the Jacobian determinant can be expressed as the sum of $S$ (total number of iterations in the sampling process) sub log-Jacobian determinants, with each term corresponding to the transformation in one iteration. In each iteration, the transformation is given by,

$$x_{s-1} = \sqrt{\alpha_{s-1}}\underbrace{\left(\frac{x_s - \sqrt{1-\alpha_s}\epsilon_\theta^{(s)}(x_s)}{\sqrt{\alpha_s}}\right)}_{} + \underbrace{\sqrt{1-\alpha_{s-1}}\cdot\epsilon_\theta^{(s)}(x_s)}_{}, \tag{30}$$

for a sequence of $\alpha_s \in (0,1)$. To compute the Jacobian determinant for this transformation, it suffices to compute the Jacobian of $\epsilon_\theta^{(s)}$. Since $\epsilon_\theta^{(s)}$ is parameterized by a U-Net architecture, its Jacobian can be directly read from the trained U-Net.

The reverse of the sampling process, which encodes a test image into latent noise, can be achieved by simulating the reverse of an ODE. In fact, DDIM can be considered as an Euler method to solve ODEs. Specifically, the iterative formula can be represented as

$$\sqrt{\frac{1}{\bar{\alpha}_{s-1}}}x_{s-1} - \sqrt{\frac{1}{\bar{\alpha}_s}}x_s = \left(\sqrt{\frac{1}{\bar{\alpha}_{s-1}}} - \sqrt{\frac{1}{\bar{\alpha}_s}}\right)\epsilon_\theta(x_s, s), \tag{31}$$

We set $y_s := \sqrt{\frac{1}{\bar{\alpha}_s}}x_s$ and $p_s := \sqrt{\frac{1}{\bar{\alpha}_s}} - 1$,

$$y_{s-1} - y_s = (p_{s-1} - p_s)\epsilon_\theta(x_s, s). \tag{32}$$

In the limit of small steps, this equation becomes an ODE:

$$dy_s = \epsilon_\theta(x_s, s)dp_s. \tag{33}$$

Then, the reversal of this ODE can be derived as follows:

$$y_{s+1} - y_s = (p_{s+1} - p_s)\epsilon_\theta(x_s, s), \tag{34}$$

which becomes,

$$\sqrt{\frac{1}{\bar{\alpha}_{s+1}}}x_{s+1} - \sqrt{\frac{1}{\bar{\alpha}_s}}x_s = \left(\sqrt{\frac{1}{\bar{\alpha}_{s+1}}} - \sqrt{\frac{1}{\bar{\alpha}_s}}\right)\epsilon_\theta(x_s, s). \tag{35}$$

**Remark 3.** Extending our evaluation method to generative models with a non-deterministic reverse process, such as the Denoising Diffusion Probabilistic Models (DDPM) (Ho et al., 2020), would be an intriguing direction for future research.

## D  Additional empirical results

### D.1  Additional experiments for comparing CLT and EEs

In this section, we also design more complicated data generation process to compare the inference performances between our proposed EEs methods our proposed CLT ($Z$-statistics and $T$-statistics in Theorem 3.1 and Corollary 3.1, respectively) and EEs ($Z$-statistics and $T$-statistics in Theorem 4.1 and (B.1), respectively). Beyond the linear transformation of normal input (13), we also consider 5 types of input distribution of $X, X_1, X_2$: normal, skew-normal, $\chi^2$, Gamma and Beta distributions. The output data generation mechanism follows:

$$Y_1 = A \odot g(X_1) + B, \qquad Y_2 = (A + \epsilon I_d) \odot g(X_2) + B + \epsilon, \tag{36}$$

where $\odot$ denotes the element-wise product, $g(\cdot)$ is the transformation function with certain smoothness to control the non-Gaussian degree of outputs, closer to the generative models in certain sense. Consider the following 8 forms of $g(\cdot)$ for evaluations: (1) $g(X) = X$, (2) $g(X) = X^2$, (3) $g(X) = |X|^{3/2}$, (4) $g(X) = \sin(2\pi x)$, (5) $g(X) = \text{logit}(X)$, (6) $g(X) = \sigma(X)$ with sigmoid transformation $\sigma(z) = 1/(1+e^{-z})$, (7) $g(X) = \text{ReLu}(X) = \max(0, X)$, (8) $g(X) = \sigma(A \odot \sigma(A \odot \sigma(X) + B) + B)$ with multilayer sigmoid transformation. Other settings follow the simulated model (13). For each setting, we evaluate the performance of each method based on the following metrics: (1) empirical CI coverage rate, (2) empirical statistical power, and (3) empirical CI length.

The results of CIs based on $T$-statistics (27) with unknown variance for the linear transformation and the sigmoid transformation are presented, respectively, in Figures 5 and 6. In Figures 5 and 6, the plots in the left column show the results for the EEs method, while the plots in the right column show the results for the CLT method. Each row corresponds to one of the three inference metrics. In all Figures, the $x$-axis represents the ground truth value $\Delta = \mathbb{E}_G[\log p_1 - \log p_2]$, and a dashed line marks the nominal 90% coverage level. Figures 5 and 6 show that the CIs based on the EEs method not only can coverage the true interested parameter relative score, but also not sacrifice the testing capability and the length of CIs, yielding a better trade-off between type I error and type II error of statistical inference.

The additional simulation results of $T$-statistics under other transformations and those of $Z$-statistics are presented in Figures 7–20, respectively.

Both CLT-based and Edgeworth-expansion-based intervals are asymptotically valid. The experiment results show that when the sample size is large ($n \geq 50$), and the output distribution has a normal tail, intervals constructed from two methods are almost the same; CLT-based methods would be easier to use and understand. When sample sizes are small ($n \leq 50$), the output distribution has a heavy tail, non-negligible skewness and kurtosis, CLT-based intervals could have lower coverage rates. When the valid coverage rate (or type I error) is the main concern, Edgeworth-expansion-based methods should be preferred. If the interval length is the main concern and slightly miscoverage is acceptable, CLT-based methods will be preferred. Overall, we recommend the users to choose the CLT-based statistics and intervals when the sample size $n \geq 50$ and the output distribution has a normal tail; and to choose the Edgeworth-expansion-based ones when $n \leq 50$ or the output distribution is far away from normal.

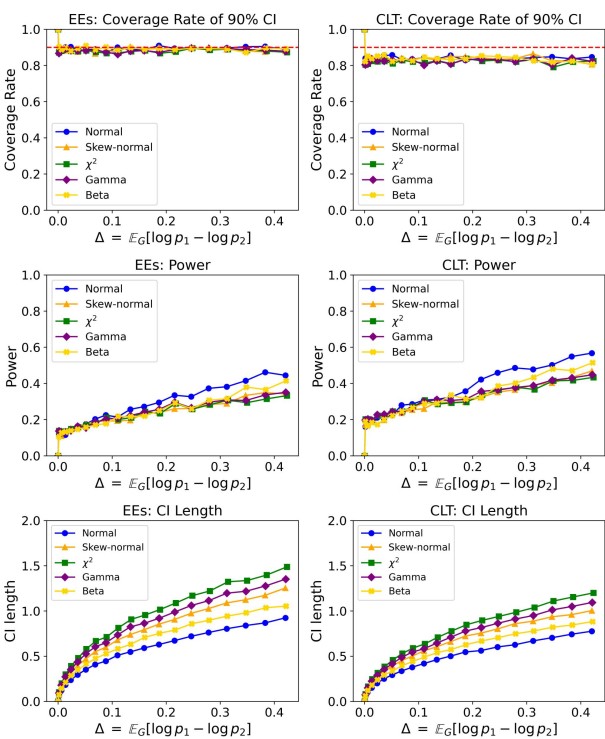

Figure 5: Coverage rate, power and length of CIs constructed by EEs and CLT of $T$-statistics under linear transformation.

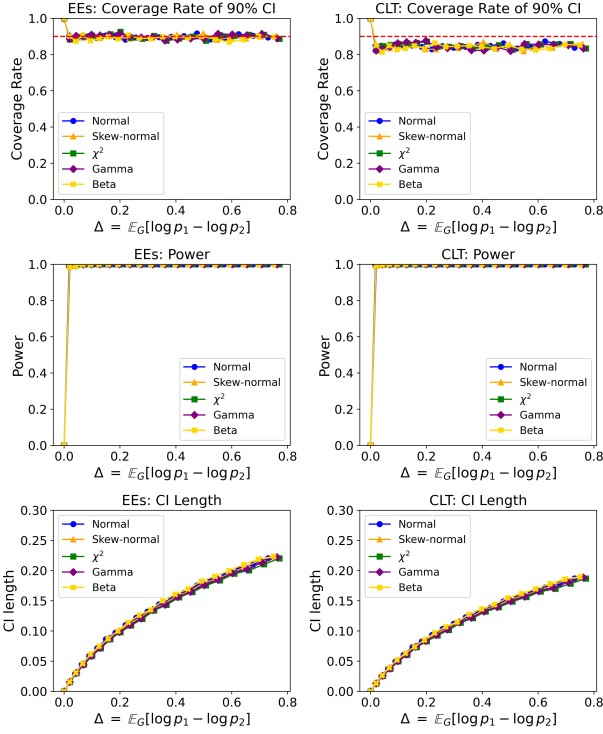

Figure 6: Coverage rate, power and length of CIs constructed by EEs and CLT of $T$-statistics under sigmoid transformation.

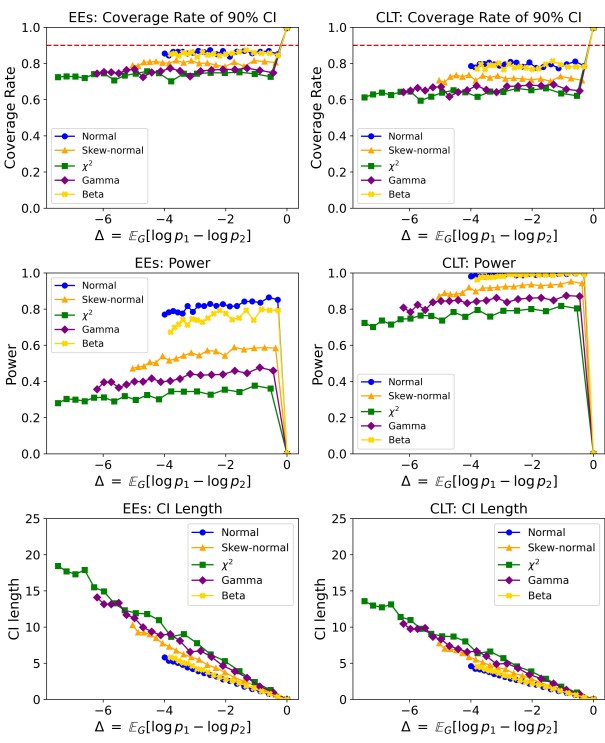

Figure 7: Coverage rate, power and length of CIs constructed by EEs and CLT of $T$-statistics under transformation $g(X) = X^2$.

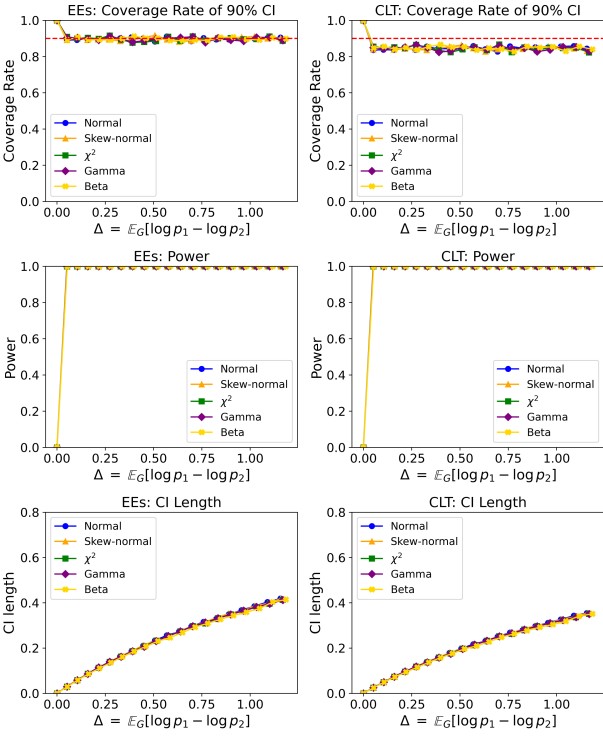

Figure 8: Coverage rate, power and length of CIs constructed by EEs and CLT of $T$-statistics under transformation $g(X) = sin(2\pi X)$.

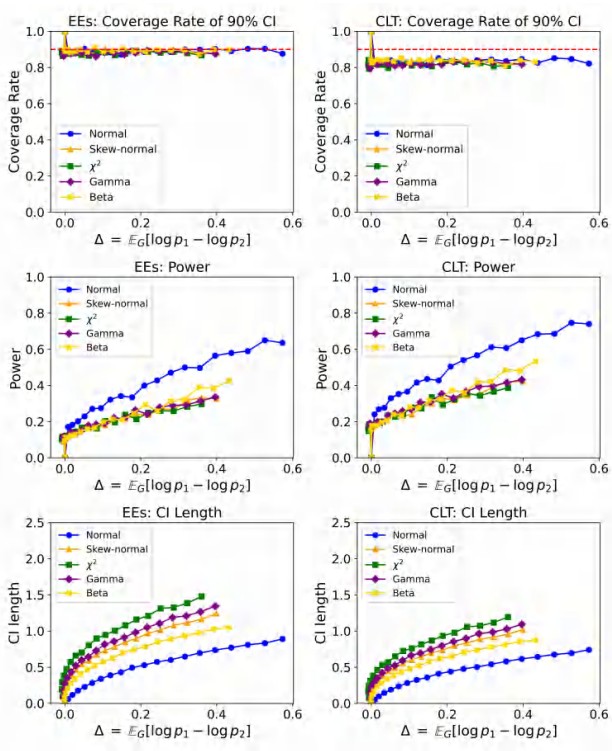

Figure 9: Coverage rate, power and length of CIs constructed by EEs and CLT of $T$-statistics under ReLU transformation $g(X) = \mathrm{ReLu}(X) = \max(0, X)$.

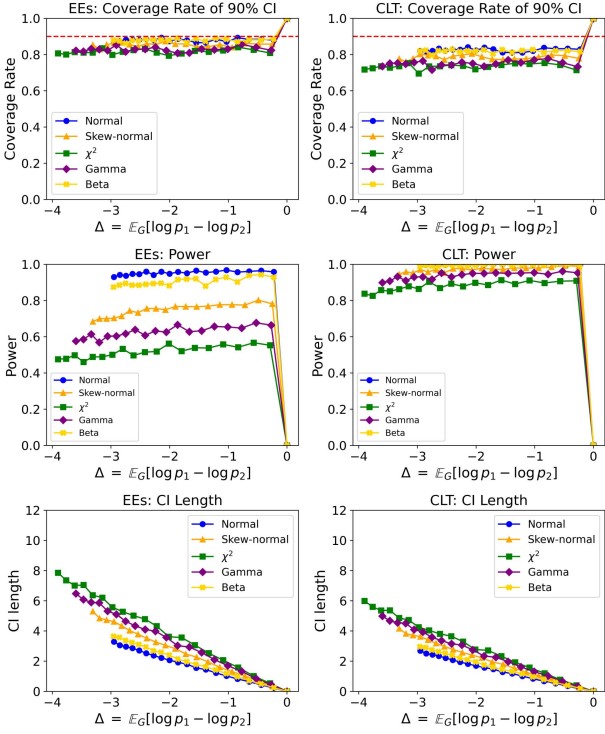

Figure 10: Coverage rate, power and length of CIs constructed by EEs and CLT of $T$-statistics under Logit transformation $g(X) = \mathrm{logit}(X)$.

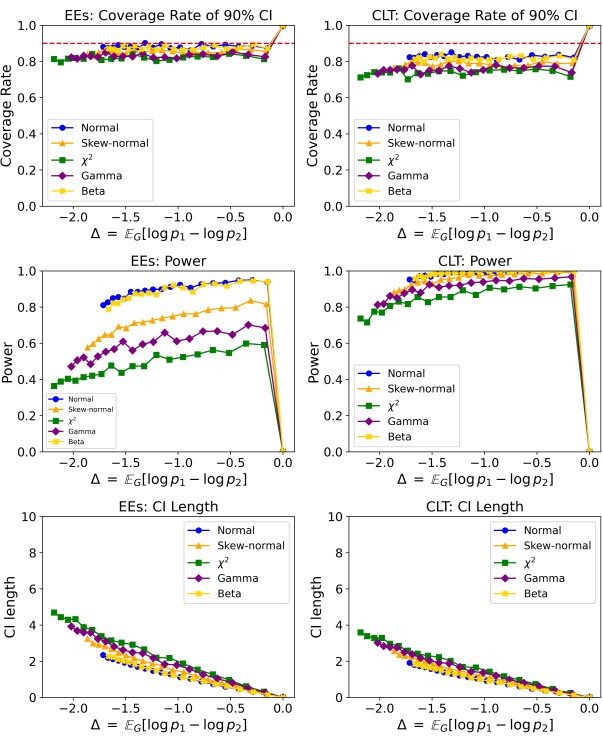

Figure 11: Coverage rate, power and length of CIs constructed by EEs and CLT of $T$-statistics under transformation $g(X) = |X|^{3/2}$.

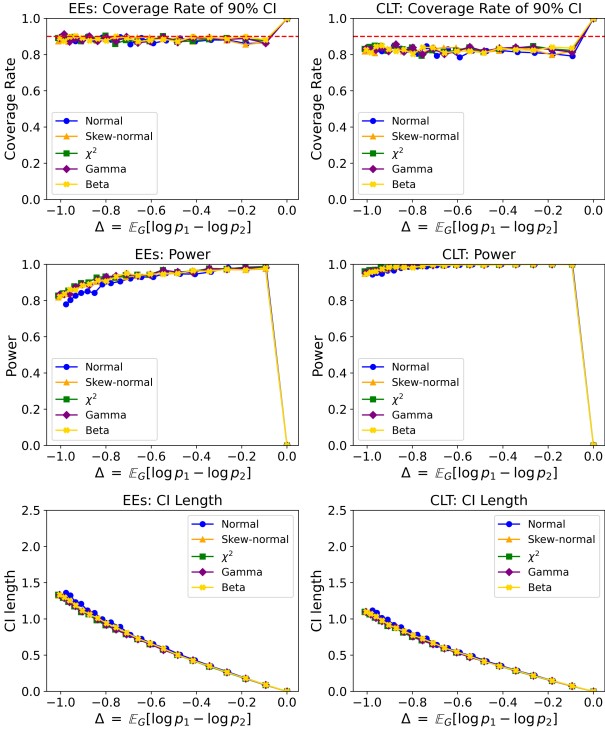

Figure 12: Coverage rate, power and length of CIs constructed by EEs and CLT of $T$-statistics under multilayer sigmoid transformation $g(X) = \sigma(A \odot \sigma(A \odot \sigma(X) + B) + B)$.

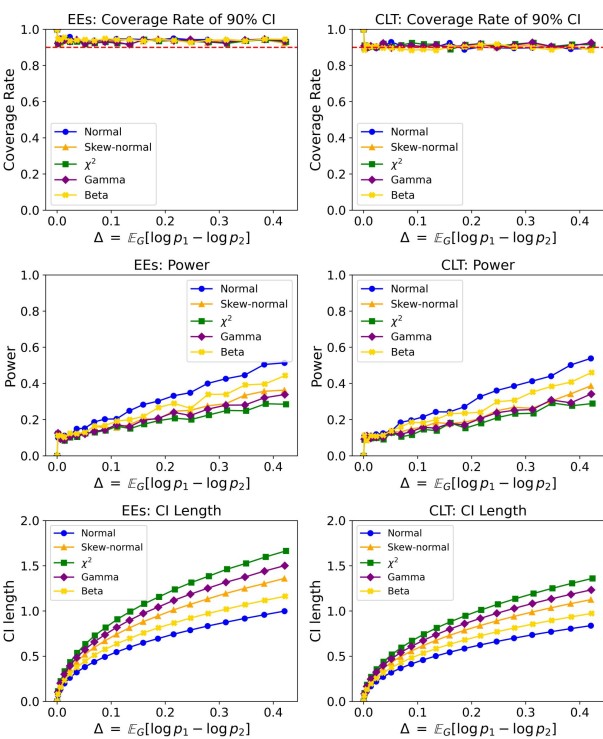

Figure 13: Coverage rate, power and length of CIs constructed by EEs and CLT of $Z$-statistics under linear transformation.

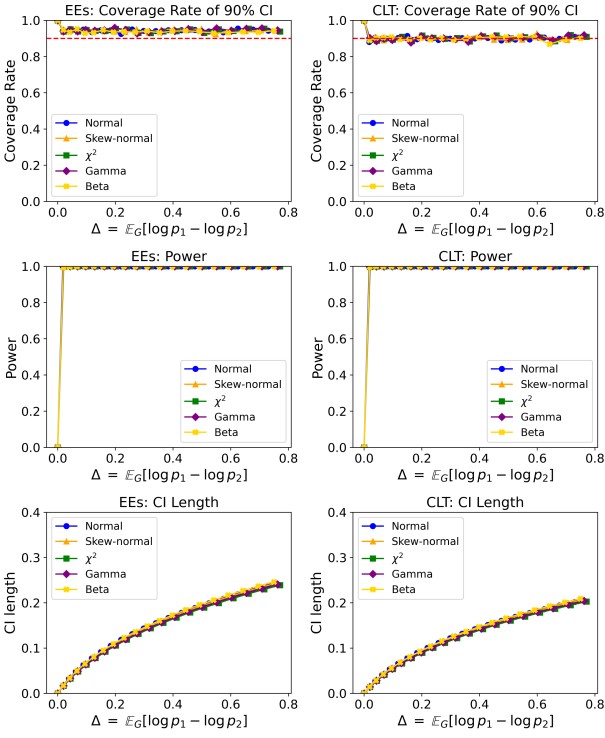

Figure 14: Coverage rate, power and length of CIs constructed by EEs and CLT of $Z$-statistics under sigmoid transformation.

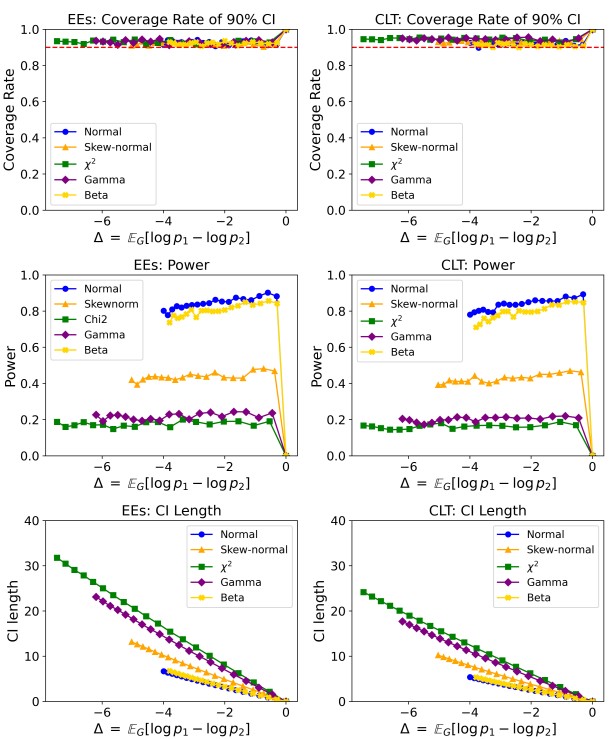

Figure 15: Coverage rate, power and length of CIs constructed by EEs and CLT of $Z$-statistics under transformation $g(X) = X^2$.

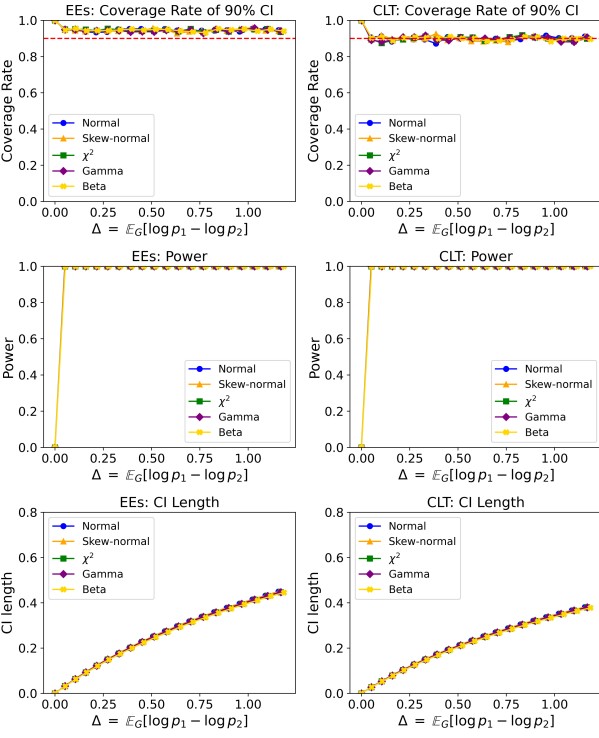

Figure 16: Coverage rate, power and length of CIs constructed by EEs and CLT of $Z$-statistics under transformation $g(X) = sin(2\pi X)$.

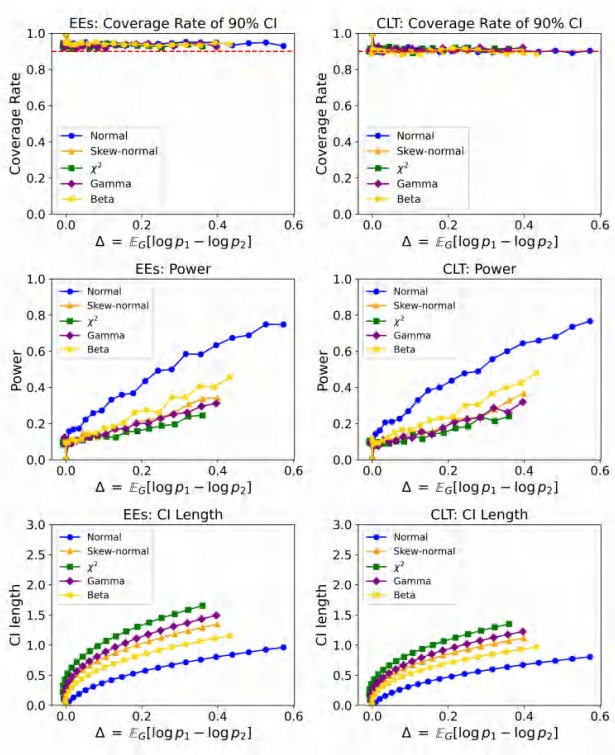

Figure 17: Coverage rate, power and length of CIs constructed by EEs and CLT of $Z$-statistics under ReLU transformation $g(X) = \mathrm{ReLu}(X) = \max(0, X)$.

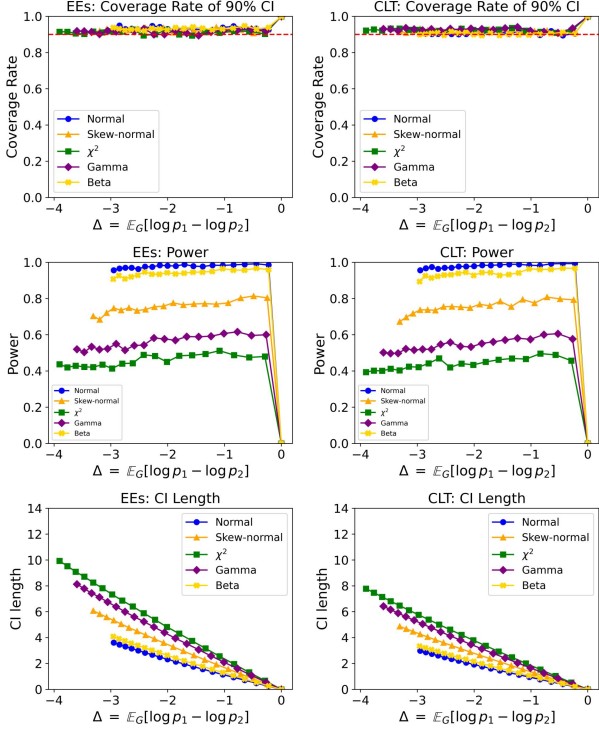

Figure 18: Coverage rate, power and length of CIs constructed by EEs and CLT of $Z$-statistics under Logit transformation $g(X) = \mathrm{logit}(X)$.

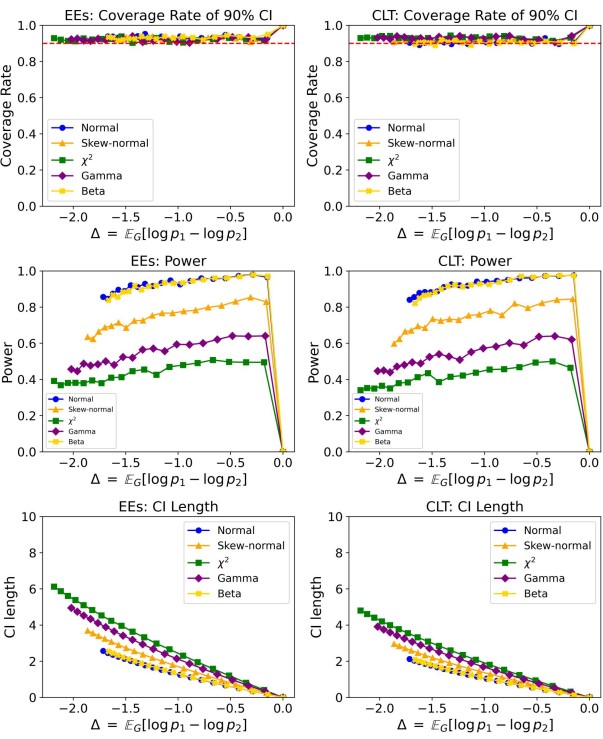

Figure 19: Coverage rate, power and length of CIs constructed by EEs and CLT of $Z$-statistics under transformation $g(X) = |X|^{3/2}$.

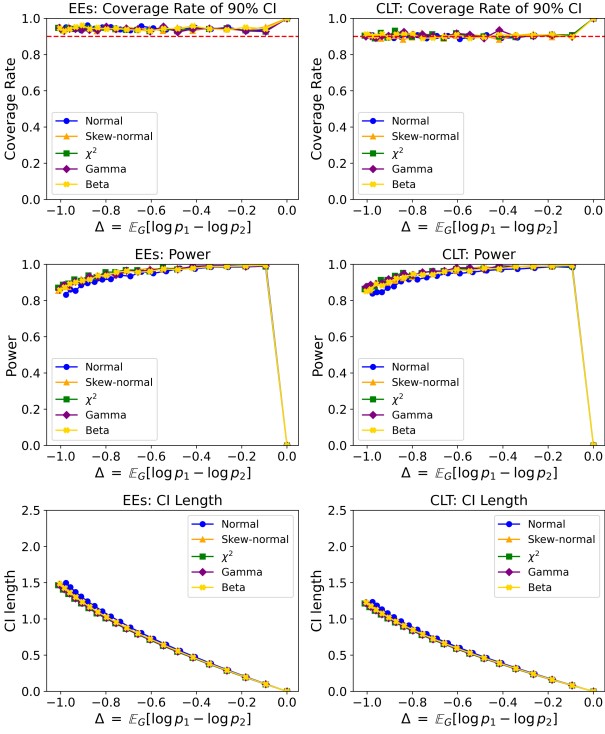

Figure 20: Coverage rate, power and length of CIs constructed by EEs and CLT of $Z$-statistics under multilayer sigmoid transformation $g(X) = \sigma(A \odot \sigma(A \odot \sigma(X) + B) + B)$.

### D.1.1 Choices of Optimal $n$ Using EEs

Figures 22 and 21 compare the inference performances of EEs of $T$-statistics with unknown variance across different input distributions of $X$ and different sample sizes $n$, and we use the CLT of $T$-statistics as competitor. We find that EEs successfully keep the coverage rate at the nominal level while CLT suffers bigger coverage errors over $\alpha$, especially when the sample size $n$ is relatively small (like $n < 100$). These further point out that the EEs can be used for early stopping and the optimal training sample determination in the inference of relative scores.

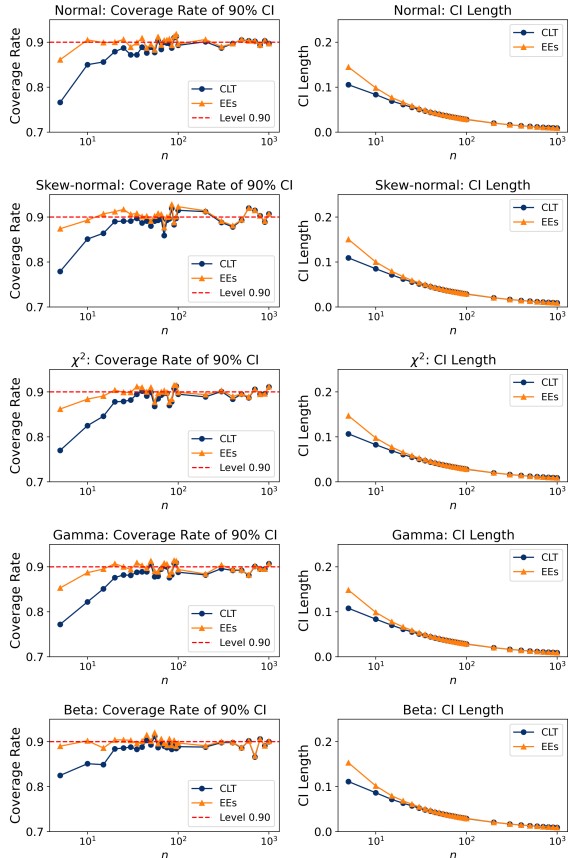

Figure 21: Coverage rates and length of confidence intervals obtained by EEs (12) and CLT (7) across different distributions of $X$ and sample sizes $n$ using simulated data generated by (36) with sigmoid transformation, $\epsilon = 0.07$.

## D.2 Additional details for Section 5.1.1

We investigate why existing estimators fail to provide valid confidence intervals by examining their distributions numerically. Figure 23 presents histograms of the estimators alongside the true values of relative KL divergence and relative $W_2$ distance when $\epsilon = 0.05$. The results show that these estimators exhibit significant bias, making them unsuitable for reliable inference. This empirical observation is consistent with our discussion of the challenges in estimating KL divergence and $W_2$ distance (Section 3).

We also provide more details on the estimated inverse learned by auto-encoders used in the simulations of Section 5.1.1. Specifically, we consider the same data generation mechanism (13), and generate $N = 5 \times 10^5$ samples from $Y_1$ (from $g_1$) and $Y_2$ (from $g_2$) and train two auto-encoders to reconstruct the data. Both the encoder and decoder are fully connected neural networks with a single hidden layer containing 100 units. The auto-encoders are trained for 20 epochs using the Adam optimizer (Kingma, 2014). To enforce that the

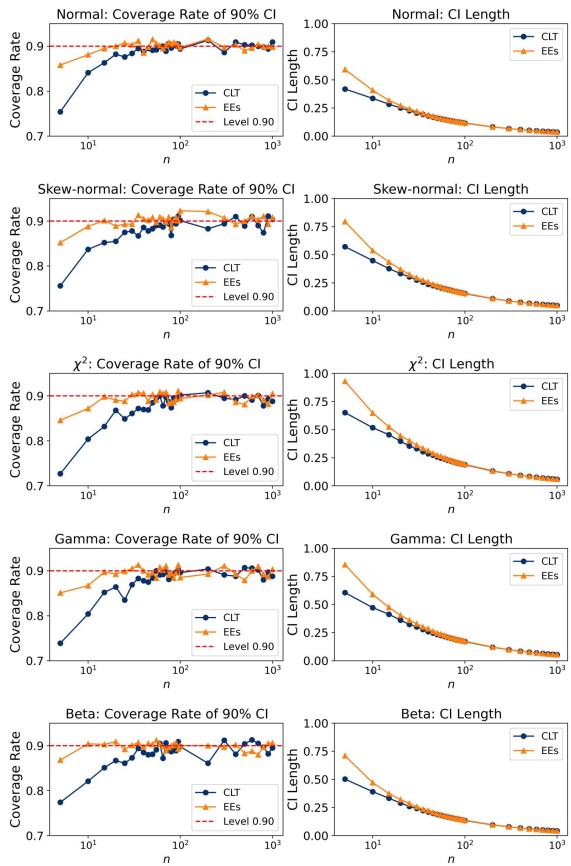

Figure 22: Coverage rates and length of confidence intervals obtained by EEs (12) and CLT (7) across different distributions of $X$ and sample sizes $n$ using simulated data generated by (36) with linear transformation, $\epsilon = 0.07$.

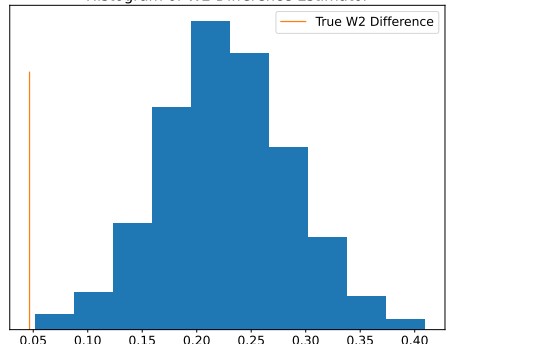

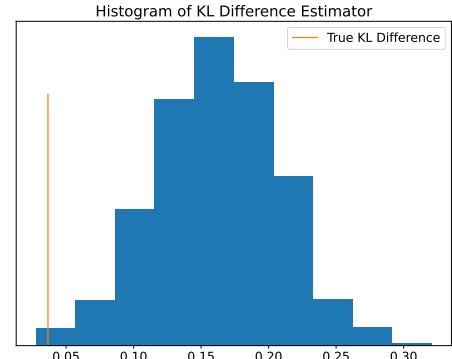

Figure 23: Histogram of existing estimators for the $W_2$ distance and KL divergence. The vertical line indicates the true value in the simulated example.

encoder maps the data to $\mathcal{N}(0,1)$, we introduce a penalty term $\text{KL}(\mathcal{N}(\hat{\mu}, \hat{\Sigma}) \| \mathcal{N}(0, I_d))$, where $\hat{\mu}$ and $\hat{\Sigma}$ are the empirical mean and covariance matrix of the encoder outputs. We use the two learned encoders as $g_1^{-1}$ and $g_2^{-1}$, respectively.

To assess the performance of our method, we again generate $n = 1000$ samples from $Y$, compute the estimator in (29), and construct the confidence intervals using (7). The coverage rates over 1000 repeated experiments are summarized in Figure 3 (denoted as "Ours (Auto Encoder)"). The results show that our method achieves coverage rates close to the target level, even when using the learned inverse functions.

More generally, for the case where the generator $g$'s inverse $g^{-1}$ is not directly available, we can train an auto-encoder minimizing the reconstruction error using the data generated from $g$ and use the encoder as $g^{-1}$. To understand the effect of the approximation errors in $g^{-1}$, we conduct a sensitivity analysis using the simulated data. We compose the encoder for $Y_2$ with a perturbation function

$$h(X) = X \circ (1 + \delta Z),$$

where $Z \sim \mathcal{N}(0, I)$ is a gaussian vector with the same shape as $X$, $\circ$ denote the elementwise product. $\delta$ controls the strength of the perturbation. We compute the likelihood of $Y_2$ using the perturbed encoder and evaluate the coverage rate of our method. The results are given in Table 4. In general, when the approximation error is small (in this case, $\delta \leq 0.01$), the coverage is not substantially affected, indicating that our approach is not highly sensitive to small approximation error.

Table 4: Coverage rate and bias under different perturbation levels

| $\delta$ | 0.00 | 0.01 | 0.02 | 0.03 | 0.04 | 0.05 |
|---|---|---|---|---|---|---|
| Coverage Rate | 0.906 | 0.894 | 0.842 | 0.770 | 0.660 | 0.462 |
| Bias | $3.74 \times 10^{-3}$ | $4.79 \times 10^{-3}$ | $7.57 \times 10^{-3}$ | $1.38 \times 10^{-2}$ | $1.94 \times 10^{-2}$ | $2.88 \times 10^{-2}$ |

### D.3 Additional details for Section 5.1.3

For wikitext data, there might exist a potential non-iid structure on the datasets of language models, when considering that the dataset may contain multiple paragraphs from the same article, such as the WikiText dataset. In such cases, the sequence of the input data $\{X_i\}_{i \geq 1}$ (random variables) can be said to have a $m$-dependent structure: there exists a small integer $m$ (i.e., the maximum of the number of paragraphs in each article), such that for any $n \geq 1$ and $k \geq m$, it holds that $X_{n+k}$ is independent of $\mathcal{F}_n = \sigma(X_1, \ldots, X_n)$, the $\sigma$-field generated by $\{X_i\}_{i=1}^n$. In such $m$-dependent case, under two mild conditions: (C1) $\{X_i\}_{i=1}^n$ have uniformly bounded variance such that $\{mn^{1/3}\}^{-1}\sqrt{\text{Var}(\sum_{i=1}^n X_i)} \to \infty$ as $n \to \infty$, and (C2) $m = o(n^{1/3})$, our proposed estimator of the relative score is still unbiased and asymptotically normal, and hence the statistical inference and uncertainty quantity based on the relative score still work. The corresponding theoretical results about the CLT and EEs for m-dependent sequences can be found in Van der Vaart (2000) and Hall (2013).

We provide experiments comparing CLT-based intervals and EEs intervals. Figures 24 presents the pairwise CIs comparison results of GPT2 vs. GPT2-Large.

### D.4 Additional details for Section 5.2

For TriviaQA, we compute the likelihood for a single answer per data point. Specifically, we use the answer string provided by example["answer"]["value"] and do not perform additional answer normalization or aggregation of probability mass across multiple acceptable answers. The likelihood is therefore evaluated on this single reference answer only.

In addtion, we provide experiments comparing CLT-based intervals and EEs intervals. Figures 25 and 26 present the pairwise CIs comparison results of LLMs: OPT-1.3B vs. Mistral-7B, and OPT-1.3B vs. Llama3-8B. The results indicate that both of the Mistral-7B and the Llama3-8B models are significantly different from OPT-1.3B model, which makes sense because of the quite different memory sizes.

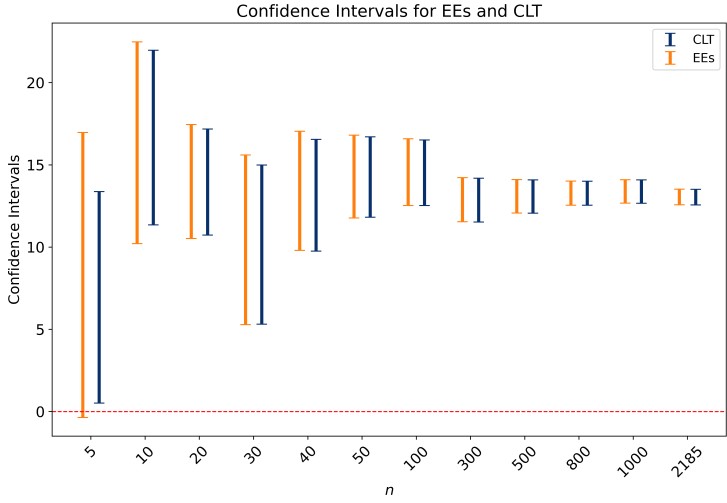

Figure 24: Comparison between GPT2 and GPT2-Large on WikiText-2 with different sample size

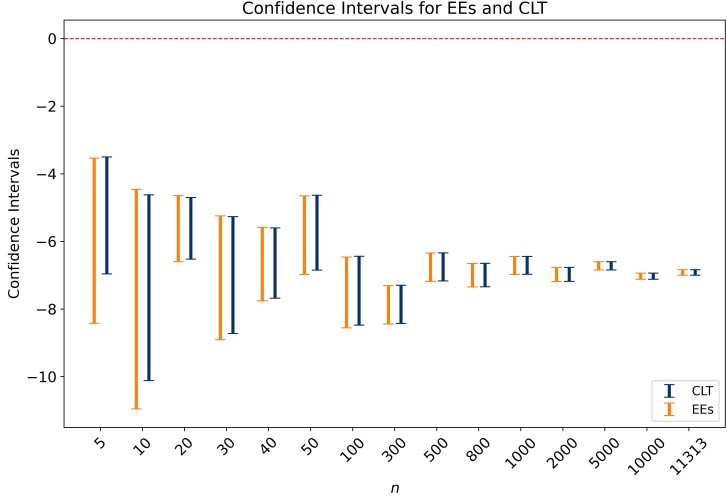

Figure 25: CIs of OPT-1.3B vs. Mistral-7B for various sample size $n$.

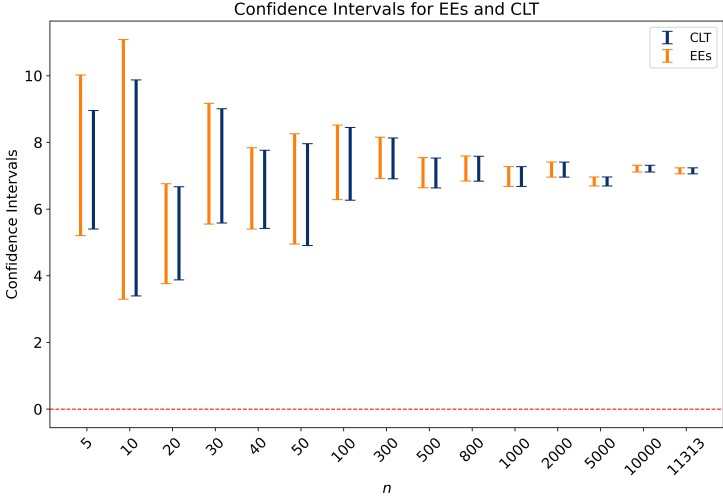

Figure 26: CIs of Llama3-8B vs. OPT-1.3B for various sample size $n$.

## D.5 Additional Experiments on Generative Models for Image

We apply our method to evaluate generative models on the CelebA dataset (Liu et al., 2015). Specifically, we trained a VAE model using the default settings from the GitHub repository[13] and compared it with a pre-trained DDIM model (Song et al., 2021) with $S = 20$ denoising steps. The results are summarized in Table 5. Our results are consistent with FID scores, but our method additionally provides statistical confidence in the comparisons.

Table 5: Comparison of a VAE model and a DDIM model with $S = 20$ denoising steps on CelebA data. Our confidence interval doesn't cover 0, indicating the DDIM model is significantly better than the VAE model.

| Model $M$ | FID | $\widehat{\text{CI}}$ of $\delta(\hat{\mathbb{P}}_M, \hat{\mathbb{P}}_{\text{DDIM}_{20}})$ |
|---|---|---|
| VAE | 150.65 | (-45912.80, -45690.22) |
| DDIM$_{20}$ | 15.26 | - |

## E Future Directions

We outline several promising directions for future research.

- Extension to the comparison of multiple generative models. Our estimator (5) of the relative score and its asymptotic distribution characterization Theorem 3.1 naturally extend to pairwise comparisons of multiple generative models. Combined with Fan et al. (2024), we can identify the best-performing model and establish the full ranking of multiple generative models with statistical confidence.

- Heterogeneity in relative performance. There may be significant heterogeneity in the relative performance of generative models across test datasets, e.g., a model that excels at generating cat images might perform poorly on car images. In future works, we aim to use our method to identify the strengths of different generative models with statistical confidence, which can further guide the development of expert models that strategically leverage the strengths of individual models for improved performance.

---

[13]https://github.com/AntixK/PyTorch-VAE

