# OpenReview forum: "Statistical Inference for Generative Model Comparison"
_TMLR — Accepted by TMLR_

### Review · Reviewer_ayqR · 2025-12-01

**Summary Of Contributions:**

To address the "lacking uncertain property" of metrics like FID or kernel-based metrics, the paper proposes a relative KL divergence metric and extends the metric to conditional generative models. Theorems and experiments are presented.

**Additional Comments:**

1. For the theorem, I do not rigorously check the correctness.

2. Better presentations and illustrations of the motivation, the method are expected.

**Audience:**

Yes

**Audience Explanation:**

The topic "reasonable metric of generative models" is very vital in the community. The community indeed needs better and more reasonable metrics.

**Claims And Evidence:**

No

**Claims Explanation:**

1. The paper challenges the "lacking uncertain property" of FID or other metrics. However, for readers even working on generative models, we cannot get the point "uncertain property". Why does the proposed metric (related KL divergence) formulate the uncertain property? Please refer to the section "Requested changes".

2. From the experiments, it is somehow hard to find the superiority of the proposed metric. Better presentation is expected.

**Requested Changes:**

1. The motivation. What is the "lacking uncertain property" of previous metrics? Can authors provide more thorough and convincing illustrations? I guess, authors may challenge the "ground truth" (i.e., the statistical properties of the training dataset) is statically calculated, which cannot reflect the property of the distribution. That is the flaw of the statically constructed dataset (e.g., ImageNet). If we check the calculation of FID, the mean $\mu$ and std $\sigma$ of high level Inception features of the training dataset are used. It is about static property, rather than "lacking uncertain property". Moreover, why does the proposed metric address the "lacking uncertain property" of previous metrics?

2. The calculation of the proposed metric. In Eq. (3), why do authors claim that calculating the entropy of the training/real dataset $\mathbb{P}$ is challenging? Do authors mean that the coverage of all samples in $\mathbb{P}$ is challenging? In Eq. (3), how do authors calculate/approximate the remaining terms? I mean, the integration over $\mathbb{P}$ is also challenging. In practice, we do not know the probability of $y\sim \mathbb{P}(y)$. In most cases, we use $\frac{1}{N}$, where $N$ is the size of sampling.

3. The results. From the presented results, I cannot see the potential value of the proposed metric.  Can the proposed metric guide a better generative model, e.g., Text-to-image models or Noise-to-image models? Figure 1 is expected to show more direct advantages of the proposed metric.

---

> ### Author Response · Authors · 2026-01-21
> **Rebuttal by Authors**
>
> We highly appreciate the reviewer's summary and comments.
>
> ### Comment 1
> > The motivation. What is the "lacking uncertain property" of previous metrics? Can authors provide more thorough and convincing illustrations? I guess, authors may challenge the "ground truth" (i.e., the statistical properties of the training dataset) is statically calculated, which cannot reflect the property of the distribution. That is the flaw of the statically constructed dataset (e.g., ImageNet). If we check the calculation of FID, the mean $\mu$ and std $\sigma$ of high level Inception features of the training dataset are used. It is about static property, rather than "lacking uncertain property". Moreover, why does the proposed metric address the "lacking uncertain property" of previous metrics?
>
> ### Response
> First, FID score is usually based on a fixed number of real images, it could be a randomly drawn test sample rather than the entire dataset. As a result, evaluation metrics are subject to randomness in the test sample drawn, and different test draws can lead to different metric values. Second, even if we use a fixed test data set as a representation of real data, FID still requires sampling from the generative model, which introduces additional variability. For instance, on CIFAR-10, sampling $10,000$ images from the VAE model in Table 1 yields a mean FID of $175.57$ with a standard deviation of $0.34$ (computed from 100 repeats). Together, these sources of randomness imply that when some metric, e.g., FID, of some model exceeds that of another model, the observed difference may be driven by sampling variability rather than a genuine performance gap between the models.
>
> For our metric, we explicitly quantify how much the estimated value can deviate from its population (true) value due to the randomness discussed above, i.e., performing uncertainty quantification. This allows us to rule out conclusions that one model outperforms another purely by chance. In contrast, for commonly used metrics such as FID, the sampling distribution is difficult to characterize because the metric itself is significantly biased: the expected value of the metric computed from a finite sample is systematically different from the true population value (FID score computes the Wasserstein distance between two Gaussian distributions at the end, as shown in Figure 4, even for Gaussian distributions, the empirical estimator is highly biased ). Our metric admits an unbiased estimator due to a key cancellation, which in turn enables the uncertainty quantification described above.
>
> In addition, it would be very helpful if the reviewer could clarify what is meant by the ``training dataset'' in the question. We understand this as the test dataset used for model evaluation.

---

> > ### Comment · Reviewer_ayqR · 2026-01-21
> >
> > Thank the authors for their reply.
> >
> > First, authors are confused by the concept of "the training dataset" in my comments. I want to clarify that, to the best of my knowledge, "the training dataset" denotes the "ground truth" in the calculation of FID. The calculation of FID can be simply interpreted as ${\rm distance}({\rm generated\ samples}, {\rm ground\ truth\ samples})$. As I know, the total training dataset (e.g., the training dataset of ImageNet) is used as the "$\rm ground\ truth\ samples$", which is a difference between generative models and recognition models. For recognition models, we use **the test dataset** to evaluate the performance. However, for generative models, **the training dataset** is employed to evaluate the performance. And, authors mentioned that "FID score is usually based on a fixed number of real images, it could be a randomly drawn test sample rather than the entire dataset". That is a wrong concept. As I know, in computer vision, FID is calculated with **all training samples/images**. Other reviewers and editors can confirm that point. Thus, I sincerely suggest the authors re-illustrate the motivation. What is the uncertain property? What is the origination of the **bias**? I mean, the bias can originate from multiple facts: 1). the gap between the finite training samples and the distribution $P_{\rm data}$, where $P_{\rm data}$ is the target generative models want to learn. 2). Finite generated samples can also lead to bias. Finite generated samples cannot represent the distribution $P_{\theta}$ of generated samples. However, the authors claim that real images are randomly sampled from the training dataset, which is not the key.
> >
> > Second, the FID of samples of VAE varies with the repeatment. As I mentioned, if you use **the total training dataset** as the real images in the calculation of FID, there are no randomness in the calculation of FID. Using the total training dataset as the source of ground truth is commonly adopted in the community. In most open-sourced repositories, they provide the pre-calculated $\mu$ and $\sigma$ of the training dataset for the calculation of FID.

---

> > > ### Author Response · Authors · 2026-01-21
> > > **Author response to Reviewer ayqR**
> > >
> > > We appreciate the reviewer for the clarification. If we interpret FID as "${\rm distance}({\rm generated\ samples}, {\rm ground\ truth\ samples})$", we agree with the reviewer that when the entire training set is used for the "${\rm ground\ truth\ samples}$", there is no randomness associated with the "${\rm ground\ truth\ samples}$" term in the FID computation. However, we still need to generate samples for "${\rm generated\ samples}$". We would like to clarify that the randomness of the FID score of the VAE model in our rebuttal is not from random samples of the training dataset, but from randomness of samples generated from the VAE. For our VAE model in Table 1, we use the entire dataset as the real images in the calculation of FID. We repeatedly sample 10,000 images from the trained VAE model, We repeat this sampling procedure 100 times and compute FID for each repetition, which yields a mean FID of 175.57 with a standard deviation of 0.34. Under the interpretation of FID as "${\rm distance}({\rm generated\ samples}, {\rm ground\ truth\ samples})$", the calculation can still exhibit variability due to the randomness of the "${\rm generated\ samples}$".
> > >
> > > We would like to further clarify that in our framework, we view the ground truth distribution as an unknown population distribution, and the training data is viewed as a finite sample from the distribution (corresponding to the empirical approximation of $p(y)$ with $1/N$ mentioned in the reviewer's second requested change). This finite-sample approximation is the main source of the uncertainty that we quantify in our work. From this perspective, FID is viewed as the Wasserstein /  Fréchet distance between the transformed distribution induced by the Inception model.  As shown in Figure 4, even for Gaussian distributions, accurately estimating the Wasserstein distance from finite samples is challenging. Therefore, the uncertainty arising from finite "${\rm ground\ truth\ samples}$" is difficult to quantify for FID.

---

> > > > ### Comment · Reviewer_ayqR · 2026-01-23
> > > >
> > > > Thank the authors for the clarification. I get the point of "uncertain property of FID". Please add this point into the revised manuscript.
> > > >
> > > > "We repeatedly sample 10,000 images from the trained VAE model, We repeat this sampling procedure 100 times and compute FID for each repetition, which yields a mean FID of 175.57 with a standard deviation of 0.34." The repeated sampling procedure leads to the standard deviation, coinciding with the expection.

---

> ### Author Response · Authors · 2026-01-21
> **Rebuttal by Authors**
>
> ### Comment 2
> > The calculation of the proposed metric. In Eq. (3), why do authors claim that calculating the entropy of the training/real dataset $\mathbb{P}$ is challenging? Do authors mean that the coverage of all samples in $\mathbb{P}$ is challenging? In Eq. (3), how do authors calculate/approximate the remaining terms? I mean, the integration over $\mathbb{P}$ is also challenging. In practice, we do not know the probability of $y \sim \mathbb{P}(y)$. In most cases, we use $\frac{1}{N}$, where $N$ is the size of sampling.
>
> ### Response
> We agree with the reviewers that the probability distribution $\mathbb{P}$ of the test sample is typically unknown (Here, the density of the generative model is assumed to be available, as provided in many open-source models), and we do not directly compute any integrals with respect to $\mathbb{P}$. Instead, as suggested, we have access to test data points $Y \sim \mathbb{P}$ and estimate expectations using sample averages.
>
> Entropy is difficult to evaluate from finite samples because it involves the quantity $\mathbb{E}[\log p(Y)]$, where the density $p$ of the test distribution is unknown and must be estimated from the finite sample. Density estimation is an intrinsically difficult problem and is susceptible to the curse of dimensionality.
>
> For instance, when the KL divergence is estimated by some k-nearest neighbour estimator, [1] showed that the minmax rate of the expected squared error is roughly the order of $n^{-2/d}$, where $n$ is the sample size, $d$ is the dimension of the data. When the dimension $d$ is high, it is very difficult to estimate the quantity accurately.
>
> ### References
>
> [1] Zhao, Puning, and Lifeng Lai. "Minimax optimal estimation of KL divergence for continuous distributions." IEEE Transactions on Information Theory 66.12 (2020): 7787-7811.

---

> ### Author Response · Authors · 2026-01-21
> **Rebuttal by Authors**
>
> ### Comment 3
> > The results. From the presented results, I cannot see the potential value of the proposed metric. Can the proposed metric guide a better generative model, e.g., Text-to-image models or Noise-to-image models? Figure 1 is expected to show more direct advantages of the proposed metric.
>
> ### Response
> Metrics such as FID or perplexity provide only point estimates, and different models will almost always have different scores. In contrast, our method produces confidence intervals for the proposed metric, which can lead to the conclusion that no model is significantly better than another (such as the GPT-2 example, the performance of the quantized model is not significantly different from the original model based on the finite evaluation data), or claim one model is statistically significantly better. This allows us to distinguish genuine performance differences from variations driven by randomness, leading to more trustworthy and reliable model selections/decisions.
>
> In the revised version, we will modify Figure 1 to illustrate the necessity of uncertainty quantification. Specifically, we will show that even when two generative models are identical, FID may incorrectly suggest that one model outperforms the other due to randomness, whereas our approach could avoid such false conclusions.
>
> We would like to clarify that our metric is designed to guide model selection within the same modeling class, e.g., among text-to-image models or among noise-to-image models—but not across these two classes. This is because text-to-image models are conditional models, whereas noise-to-image models are unconditional, and the proposed metric is not intended to compare conditional and unconditional generative mechanisms directly.

---

### Review · Reviewer_AUbY · 2025-12-20

**Summary Of Contributions:**

This paper proposes a statistical framework for rigorously and quantitatively comparing two generative models. Existing evaluation metrics (e.g., FID and Wasserstein distance-based metrics) often lack uncertainty quantification, making it difficult to draw statistically grounded conclusions. The key contribution is an evaluation method tailored to the practical question of which of two generative models is better, equipped with confidence intervals and significance testing.

The core idea is to estimate a KL-based relative score rather than absolute scores. Due to a cancellation property specific to KL divergence, nuisance terms depending on the unknown true distribution vanish, and the relative score can be written as an expectation of a test-data log-likelihood difference. This yields a simple unbiased sample-mean (first-order U-statistic) estimator with $\sqrt{n}$-rate convergence and asymptotic normality, thereby enabling principled confidence intervals and statistical significance testing.

Section 5 validates the proposed method via simulations and real-data experiments across modalities, including image generation (CIFAR-10) and text generation / conditional evaluation (WikiText-2, TriviaQA). The resulting rankings are broadly consistent with standard metrics (e.g., FID and perplexity) while additionally quantifying statistical significance, and the experiments suggest the proposed method provides better-calibrated inference than several baselines.

**Audience:**

Yes

**Audience Explanation:**

Yes.

Many researchers will have concerns about the lack of uncertainty quantification in generative model evaluation. I think this paper presents an effective approach to address that issue.

**Broader Impact Concerns:**

The proposed method aims to make the evaluation and comparison of generative models more rigorous and is not intended to directly enhance generative capabilities. If anything, it may be beneficial in helping prevent overstated claims or misleading comparisons. Accordingly, I do not see major social concerns in this paper.

**Claims And Evidence:**

Yes

**Claims Explanation:**

YES.

**Theoretical evidence.** The KL-based relative score in Eq. (2) admits a key cancellation: the intractable entropy term induced by the unknown true distribution vanishes, so the target reduces to an expectation of the test-data log-likelihood difference as in Eq. (3). This paper further shows this cancellation is essentially specific to KL divergence (i.e., it does not generally hold for arbitrary $f$-divergences; Proposition 1). The resulting estimator is a simple sample mean (a first-order U-statistic), which is unbiased (Proposition 2) and asymptotically normal at $\sqrt{n}$-rate (Theorem 3.1). This directly enables normal-approximation confidence intervals (Eq. (7)) with asymptotically correct coverage $1-\alpha$ (Corollary 3.2). Section 4 additionally discusses practical extensions, including conditional generative models and small-sample settings.

**Empirical evidence.** In Section 5.1.1, simulations evaluate both coverage and power, supporting that the proposed inference procedure is well-calibrated in controlled settings. Sections 5.1.2–5.1.3 further validate the method on real benchmarks for image and text/conditional generation, showing conclusions consistent with standard metrics (e.g., FID and perplexity) while additionally providing statistical significance statements via confidence intervals.

**Requested Changes:**

I found the paper well written and well organized, and I do not think it requires major revisions. That said, one additional experiment could further strengthen the empirical evidence.

**Sample-size sensitivity on unconditional benchmarks (CIFAR-10, WikiText-2).** While the appendix investigates sample-size sensitivity in the conditional TriviaQA setting, a comparable analysis is missing for the unconditional benchmarks. It would be helpful to run the same type of subsampling study on CIFAR-10 and WikiText-2 and report how the CI width and the frequency of excluding zero change with the test-set size.

---

> ### Author Response · Authors · 2026-01-21
> **Rebuttal by Authors**
>
> We highly appreciate the reviewer's summary and positive comments.
>
> ### Comment 1
> > I found the paper well written and well organized, and I do not think it requires major revisions. That said, one additional experiment could further strengthen the empirical evidence.
> >
> > Sample-size sensitivity on unconditional benchmarks (CIFAR-10, WikiText-2). While the appendix investigates sample-size sensitivity in the conditional TriviaQA setting, a comparable analysis is missing for the unconditional benchmarks. It would be helpful to run the same type of subsampling study on CIFAR-10 and WikiText-2 and report how the CI width and the frequency of excluding zero change with the test-set size.
>
> ### Response
> We thank the reviewer for the helpful suggestion. We have conducted a sample-size sensitivity analysis for the unconditional benchmarks (in particular, WikiText-2) and report the results in the revised version. Please refer to *"Figure: Comparison between GPT2 and GPT2-Large on WikiText-2 with different sample size"* (Figure 26 on page 35) in the updated pdf file.
>
> ### Comment 2
> > The proposed method aims to make the evaluation and comparison of generative models more rigorous and is not intended to directly enhance generative capabilities. If anything, it may be beneficial in helping prevent overstated claims or misleading comparisons. Accordingly, I do not see major social concerns in this paper.
>
> ### Response
> We completely agree with the reviewer. In the revised version, we will highlight in the discussion section that our method is intended to evaluate and select among existing generative models, rather than to assist with model training or tuning.

---

### Review · Reviewer_vp9U · 2026-01-13

**Summary Of Contributions:**

The paper proposes a likelihood-based framework to compare generative models with uncertainty quantification by using the difference in KL divergences to an unknown test distribution, defining a relative KL score. The key idea is that taking the difference cancels the intractable data entropy term, so the target reduces to an expectation under the test distribution of a known log-likelihood ratio. The estimator is simply the average per example log-likelihood difference, which enables standard asymptotic inference and a variance-based confidence interval. Empirically, the authors report improved coverage compared to KL and Wasserstein based estimators paired with resampling methods, and higher power than kernel based baselines such as MMD and KSD in simulations. They further present case studies on image models (DDIM variants, normalizing flows, and VAEs) on CIFAR-10, and on language models evaluated on WikiText2 and TriviaQA.

**Audience:**

Yes

**Audience Explanation:**

This paper addresses a clear gap: most widely used generative-model metrics (e.g., FID, perplexity) are point estimates and do not provide uncertainty quantification. The proposed approach is simple and scalable, compute a held-out average log-likelihood ratio and attach confidence intervals, and it naturally extends to conditional models. As a result, it is directly relevant to modern conditional generators and language models, and provides a principled statistical perspective on generative model evaluation that many in TMLR’s audience would find useful.

**Broader Impact Concerns:**

A main concern is that statistical significance under the proposed relative-KL metric can be over-interpreted: improved likelihood does not necessarily imply better quality, usefulness, or safety, especially if rare harmful behaviors are underrepresented in the test set.

**Claims And Evidence:**

Yes

**Claims Explanation:**

The main claims are well supported by theoretical analysis. The appendix provides proofs (or standard derivations) for the key theoretical statements, including: the KL-specific uniqueness/cancellation result, unbiasedness of the proposed estimator, asymptotic normality via standard iid CLT arguments, and the Edgeworth expansion derived through cumulants and characteristic-function expansions under explicit moment and regularity assumptions. The appendix also discusses the unknown-variance (t-statistic) refinement and links it to classical results.
However, the empirical evidence would be more convincing if the paper were more explicit about how likelihoods are computed for each model family in the reported experiments. In particular, the paper does not quantify how approximation error in likelihood evaluation (when exact likelihoods are not available or are computed approximately) propagates to bias/variance and ultimately affects confidence-interval coverage and hypothesis-testing validity. Clarifying the exact likelihood computation pipeline and providing a robustness/sensitivity analysis would strengthen the empirical support.

**Requested Changes:**

# Critical
1. Clarify and validate likelihood evaluation, especially for diffusion/DDIM models. Appendix C suggests DDIM likelihoods rely on per-step Jacobian determinants; the paper should clearly state what is actually computed in experiments and how it is made tractable.
2. For each model family, explicitly specify whether the likelihood used is exact, a bound (e.g., ELBO), or an approximation, so the comparisons are consistent and interpretable.
3. Analyze how likelihood approximation error affects inference, bias, variance, and CI coverage, ideally via a sensitivity analysis or targeted ablations.
4. Provide practical guidance on when to use CLT-based intervals versus Edgeworth-expansion intervals (e.g., based on sample size and tail behavior), so users can apply the method safely.

# To strengthen the work
1. For language-model evaluation on datasets like WikiText, discuss potential non-iid structure (e.g., within-document dependence) and add guidance for multiple pairwise comparisons (e.g., FWER/FDR control).
2. For TriviaQA, specify how answer normalization/aliasing is handled and whether probability mass is aggregated over multiple acceptable answers.
3. Although the estimator is linear in the number of test points, likelihood evaluation can dominate the cost (and may be far from cheap for diffusion); report runtimes and computational details for all evaluated models and baselines.

---

> ### Author Response · Authors · 2026-01-21
> **Rebuttal by Authors**
>
> We highly appreciate the reviewer's summary and comments.
>
> ### Comment 1
> > Clarify and validate likelihood evaluation, especially for diffusion/DDIM models. Appendix C suggests DDIM likelihoods rely on per-step Jacobian determinants; the paper should clearly state what is actually computed in experiments and how it is made tractable.
>
> ### Response
>
> *We slightly abuse notation in the rebuttal to ensure compatibility with the OpenReview rendering system, replacing subscripts by function arguments, for instance, $\\begin{aligned}  g[1]:= g_{1} \\end{aligned}$.*
>
> For DDIM models, the per-step inverse update is defined in Eq. (35) of the paper. Using the above notational convention, the update from step `s-1` to step `s` can be written as
> $$
> \\begin{aligned}
> x[s] = g[1,s]^{-1}\bigl(x[s-1]\bigr)
> = \sqrt{\frac{\alpha[s]}{\alpha[s-1]}} x[s-1] + \left(
>     \sqrt{1-\alpha[s]}
>     - \sqrt{\frac{\alpha[s]}{\alpha[s-1]}}\sqrt{1-\alpha[s-1]}
>     \right)
>    +\epsilon[\theta]^{(s-1)}(x[{s-1}]).
> \\end{aligned}
> $$
>
> Likelihood evaluation is performed using Eq. (28):
> $$
> \\begin{aligned}
> \log(\hat{p} [1] (y)) = -\|{g[1]}^{-1} (y)\|_2^2/2 - d_1 \log(\sqrt{2 \pi}) + \log\bigl( \vert J[g[1]] \vert  \big({g[1]}^{-1} (y)\big) \bigr),
> \\end{aligned}
> $$
>
> where $\\begin{aligned} {g[1]}^{-1} = {g[1,1]}^{-1}\circ \cdots \circ {g[1,S]}^{-1} \\end{aligned}$. The log-determinant term decomposes additively across steps:
> $\\begin{aligned} \log\left(|J[{g[1]}^{-1}]|\right)= \sum_{s=1}^S \log{\left(|J[{g[1,s]}^{-1}]|\right)} \end{aligned}.$
> In our experiments, we mainly consider DDIM models trained on CIFAR-10. For this setting, we compute the exact Jacobian log determinants using `torch.autograd.functional.jacobian`.
> In settings with very high image resolution or an exceptionally large number of test data points, we can consider approximations of the Jacobian. In particular, the log-determinant of the Jacobian can be efficiently approximated using Stochastic Lanczos Quadrature (SLQ) (Ubaru et al., 2017). Using the identity
> $$
> \begin{aligned}
> \log \vert \det(J) \vert
> = \tfrac{1}{2} \log \bigl( J^{\top} J \bigr)
> = \tfrac{1}{2} \mathrm{trace} \bigl( \log ( J^{\top} J ) \bigr),
> \end{aligned}
> $$
> the trace term can be estimated via stochastic trace estimation (Hutchinson, 1989):
> $$
> \begin{aligned}
> \mathrm{trace} \bigl( \log ( J^{\top} J ) \bigr)
> \approx
> \sum_{i=1}^{m}
> z[i]^{\top} \log ( J^{\top} J ) z[i],
> \end{aligned}
> $$
> with random probes $z[i] \sim N(0, I)$. Each quadratic form
> $
> \begin{aligned}
> z[i]^{\top}  \log ( J^{\top} J )  z[i]
> \end{aligned}
> $
> can be computed using Lanczos iterations (Ubaru et al., 2017), which only require matrix–vector products of the form
> $
> \begin{aligned}
> J^{\top} J z[i].
> \end{aligned}
> $
> These products can be implemented using PyTorch automatic differentiation (e.g., `torch.func.jvp` and `torch.func.vjp`), with computational cost comparable to standard backpropagation. With SLQ, the overall cost of evaluating Eq. (28) is on the same order as training the model for one epoch on the evaluation dataset.
>
> We further empirically assess the approximation error of SLQ. Using the first test image of CIFAR-10 and a pretrained DDIM model from the `pytorch_diffusion` package (https://github.com/pesser/pytorch_diffusion), we compute
> $
> \begin{aligned}
> \log \big( \vert \det ( J[g[1]^{-1}] ) \vert \big)
> \end{aligned}
> $
> exactly using `torch.autograd.functional.jacobian`, obtaining a value of 16177.51. We then apply SLQ with a single random probe and 10 Lanczos steps. Across 50 runs, SLQ yields a mean estimate of approximately 16189.15 with a standard deviation of 57.13. This demonstrates that SLQ provides a reasonably accurate approximation, even with a very small number of probes and iterations.
>
>
> ### References
> [1] Michael F Hutchinson. A stochastic estimator of the trace of the influence matrix for Laplacian smoothing splines. Communications in Statistics-Simulation and Computation, 18(3):1059-1076, 1989.
>
> [2] Shashanka Ubaru, Jie Chen, and Yousef Saad. Fast estimation of $tr(f(a))$ via stochastic Lanczos quadrature. SIAM Journal on Matrir Analysis and Applications, 38(4):1075-1099, 2017.

---

> ### Author Response · Authors · 2026-01-21
> **Rebuttal by Authors**
>
> ### Comment 2
> > For each model family, explicitly specify whether the likelihood used is exact, a bound (e.g., ELBO), or an approximation, so the comparisons are consistent and interpretable.
>
> ### Response
> *We slightly abuse notation in the rebuttal to ensure compatibility with the OpenReview rendering system, replacing subscripts by function arguments, for instance, $\\begin{aligned} g[1]:= g_{1} \\end{aligned}$.*
>
> For auto-regressive language models and normalizing flow models, the likelihood can be computed exactly.
>
> For DDIM models and auto-encoder models, we use equation (28) of the paper to compute the likelihood:
>
> $$
> \\begin{aligned}
> \log(\hat{p} [1] (y)) = -\|{g[1]}^{-1} (y)\|_2^2/2 - d_1 \log(\sqrt{2 \pi}) + \log\bigl( \vert J[g[1]] \vert  \big({g[1]}^{-1} (y)\big) \bigr),
> \\end{aligned}
> $$
>
> For these models, a closed-form expression for the inverse mapping $ \\begin{aligned}{g[1]}^{-1}\\end{aligned} $ is not available. We therefore approximate $ \\begin{aligned}{g[1]}^{-1}\\end{aligned} $ using the encoder (for autoencoders) or the forward mapping (for DDIM models).
> When the reconstruction error of the autoencoder or DDIM is small, the approximation to $ \\begin{aligned}{g[1]}^{-1}\\end{aligned} $ is expected to be accurate. Empirically, in the simulated data experiment in Section 5.1.1, we use autoencoder models to approximate the distributions of $ \\begin{aligned} Y_1 \\end{aligned}$ and $ \\begin{aligned}Y_2 \\end{aligned}$. As shown in Figure 3, despite using approximate likelihoods, the resulting confidence intervals achieve coverage rates close to the nominal level.

---

> ### Author Response · Authors · 2026-01-21
> **Rebuttal by Authors**
>
> ### Comment 3
> >Analyze how likelihood approximation error affects inference, bias, variance, and CI coverage, ideally via a sensitivity analysis or targeted ablations.
>
> ### Response
> Thank you for your suggestions. We conduct a sensitivity analysis using our simulated data. We use the same autoencoder model discussed in Section 5.1.1. We compose the encoder for $Y_2$ with a perturbation function
> $$
> \begin{aligned}
> g(X) = X \circ (1 + \delta Z),
> \end{aligned}
> $$
> where $Z \sim {N}(0, I)$ is a Gaussian vector with the same shape as $X$, and $\circ$ denotes the elementwise product. The parameter $\delta$ controls the strength of the perturbation. We compute the likelihood of $Y_2$ using the perturbed encoder and evaluate the coverage rate of our method. The results are given in the following table.
>
>
> | $\delta$ | 0.00 | 0.01 | 0.02 | 0.03 | 0.04 | 0.05 | 0.06 | 0.07 | 0.08 | 0.09 | 0.10 |
> |--------------------------|------|------|------|------|------|------|------|------|------|------|------|
> | Coverage Rate            | 0.906 | 0.894 | 0.842 | 0.770 | 0.660 | 0.462 | 0.312 | 0.116 | 0.048 | 0.008 | 0.000 |
> | Bias                     | 3.74e-03 | 4.79e-03 | 7.57e-03 | 1.38e-02 | 1.94e-02 | 2.88e-02 | 3.83e-02 | 5.13e-02 | 6.50e-02 | 8.32e-02 | 1.00e-01 |
>
> In general, when the approximation error is small (in this case, $\delta \le 0.01$), the coverage is not substantially affected, indicating that our approach can tolerate small autoencoder approximation error.

---

> ### Author Response · Authors · 2026-01-21
> **Rebuttal by Authors**
>
> ### Comment 4
> >Provide practical guidance on when to use CLT-based intervals versus Edgeworth-expansion intervals (e.g., based on sample size and tail behavior), so users can apply the method safely.
>
> ### Response
> We thank the reviewer for the thoughtful question. Both CLT-based and Edgeworth-expansion-based intervals are asymptotically valid. As shown in Figure 2 in the main text and Figures 22--23 in the Appendix, when the sample size is large ($n \geq 50$) and the output distribution has a normal tail, intervals constructed from the two methods are almost the same; CLT-based methods would be easier to use and understand. When sample sizes are small ($n \leq 50$), the output distribution has a heavy tail, non-negligible skewness and kurtosis, and when the valid coverage rate (or type I error) is the main concern, Edgeworth-expansion-based methods should be preferred. If the interval length is the main concern and slightly miscoverage is acceptable, CLT-based methods will be preferred.
>
> Overall, we recommend the users to choose the CLT-based statistics and intervals when the sample size $n \geq 50$ and the output distribution has a normal tail; and to choose the Edgeworth-expansion-based ones when $n \leq 50$ or the output distribution are far away from normal.

---

> ### Author Response · Authors · 2026-01-21
> **Rebuttal by Authors**
>
> ### Comment 5
> > For language-model evaluation on datasets like WikiText, discuss potential non-iid structure (e.g., within-document dependence) and add guidance for multiple pairwise comparisons (e.g., FWER/FDR control).
>
> ### Response
> Thank you very much for the insightful comments.  Firstly, we agree with the reviewer that there might exist a potential non-iid structure on the datasets of language models, when considering that the dataset may contain multiple paragraphs from the same article, such as the WikiText dataset. In such cases, the sequence of the input data $X_i, {i\geq 1}$ (random variables) can be said to have a $m$-dependent structure: there exists a small integer $m$ (i.e., the maximum of the number of paragraphs in each article), such that for any $n \geq 1$ and $k \geq m$, it holds that $X_{n+k}$ is independent of $\mathcal{F}_n= \sigma(X_1,\dots,X_n)$,
>
> the $\sigma$-field generated by $X_i, i=1, \dots, n$. In such $m$-dependent case, under two mild conditions: (C1) $X_i$ have uniformly bounded variance such that $\{m n^{1/3}\}^{-1}\sqrt{{\rm Var}(\sum_{i=1}^n X_i)} \to \infty$ as $n \to \infty$, and (C2) $m=o(n^{1/3})$, our proposed estimator of the relative score is still unbiased and asymptotically normal, and hence the statistical inference and uncertainty quantity based on the relative score still work. The corresponding theoretical results about the CLT and EEs for m-dependent sequence can be found in Van der Vaart (2000) and Hall (2013), respectively.
>
> Secondly, when considering the multiple pairwise comparisons among different generative models, the global type I error rate control is necessary, and the guidance can be added as follows. Since that the number of candidate models $K$ is moderate (always several to ten several) and the potential dependence structure of different model pairs, we suggest the absolute criterion measure, family-wise error rate (FWER), for consideration. Formally, we can use the Bonferroni correction method to control FWER, that is, we denote ${\rm H}_{0k}$ as the null hypothesis of a pair of the model comparison, based on the constructed testing statistics and the asymptotic results of CLT/EEs, we can obtain the p-value $p_k$, $k \in [K]$. Then, for a pre-specified confidence level $\alpha \in [0,1]$, we make the statistical decision that $1(p_k \leq \alpha/K)$ for each hypothesis.
>
>
> ### Comment 6
> > For TriviaQA, specify how answer normalization/aliasing is handled and whether probability mass is aggregated over multiple acceptable answers.
>
> ### Response
> For TriviaQA, we compute the likelihood for a single answer per data point. Specifically, we use the answer string provided by example["answer"]["value"] and do not perform additional answer normalization or aggregation of probability mass across multiple acceptable answers. The likelihood is therefore evaluated on this single reference answer only. Concretely, the input formatting used in our experiments is as follows:
>     ```
>     question = data["question"]
>     ```,
>     ```
>     answer = f" {data["answer"]["value"]}"
>     ```,
>     ```
>     prompt = f"Question: {question}\nAnswer:"
>     ```
> We will clarify this preprocessing choice in the revised manuscript.
>
> ### Comment 7
> > Although the estimator is linear in the number of test points, likelihood evaluation can dominate the cost (and may be far from cheap for diffusion); report runtimes and computational details for all evaluated models and baselines.
>
> ### Response
> For Normalizing flow models and auto-regressive language models. The likelihoods $p(y)$ can be evaluated directly. Consequently, the computational cost of our method is comparable to, and often lower than, standard evaluation metrics that require sampling from the model.
>
> On the TriviaQA dataset, our method evaluates likelihoods on the ground-truth answer tokens. For the $F_1$ score, we generate the answer via greedy decoding. We evaluate the opt-1.3B model with batch size 1 on the TriviaQA validation set (11,313 questions) using a single A100 GPU. Under this setup: likelihood evaluation for our method takes 143.22 seconds. Greedy decoding and $F_1$ score evaluation take 2,799.04 seconds.
>
> For diffusion models, we agree with the reviewer that likelihood evaluation can be the dominant computational cost. As discussed in our response to Requested Change 1, when the Jacobian log-determinant is approximated using Stochastic Lanczos Quadrature (SLQ), the computational cost of likelihood evaluation is reduced substantially. In this case, the overall cost of our method is on the same order as training the diffusion model for a single epoch, making it practical for moderate-scale evaluation. We will add these computational details to the revised manuscript to improve transparency.
>
> ### References
>
> [1] Van der Vaart, A. W. (2000). Asymptotic statistics. Cambridge university press.
>
> [2] Hall, P. (2013). The bootstrap and Edgeworth expansion. Springer Science \& Business Media.

---

> ### Author Response · Authors · 2026-01-21
> **Rebuttal by Authors**
>
> ### Broader Impact Concerns:
> > A main concern is that statistical significance under the proposed relative-KL metric can be over-interpreted: improved likelihood does not necessarily imply better quality, usefulness, or safety, especially if rare harmful behaviors are underrepresented in the test set.
>
> ### Response
> We agree with the reviewer that the statistical significance under the proposed relative-KL metric should not be over-interpreted. The comparison only concerns whether the distribution of one generative model is closer to the true data distribution than the distribution of another generative model in terms of the KL-divergence. It doesn't directly imply other properties such as safety. When safety and harmful behaviors are the main concerns, specific metrics should be designed and our method can serve as a complementary metric. We will add a discussion in the revision.

---

### Public Comment · ~Sandeep_Neela1 · 2025-11-28
**Volunteer to Review**

I’d like to volunteer to review this paper. I’ve been working extensively on generative models and statistical evaluation methods, and this topic is closely related to my current work. I’m comfortable reviewing both the theoretical and experimental parts, including statistical assumptions, model comparison, and generalization of the proposed method.

I believe I can provide a useful and constructive review. I also confirm that I have no conflicts of interest with the authors.

---

### Author Response · Authors · 2026-01-27
**Summary of Changes in the Revised Manuscript**

We have uploaded a revised version of the manuscript in response to the reviewers’ comments, highlighted in red. Below, we briefly summarize the main changes and additions.


- **Clarification of likelihood evaluation**:
We clarified how likelihoods are computed for different model families (autoregressive models, normalizing flows, autoencoders, and DDIM models), including a more explicit description of the possible approximations of the log Jacobian determinant to make the computation tractable.


- **Sensitivity analysis for likelihood approximation error**:
We added a new sensitivity analysis on simulated data to study how likelihood approximation error affects inference, including CI coverage and bias, by introducing controlled perturbations to the encoder.


- **Practical guidance on inference procedures**:
We expanded the discussion comparing CLT-based and Edgeworth-expansion-based intervals, providing concrete recommendations based on sample size and tail behavior.


- **Additional empirical analyses**:
We added sample-size sensitivity analyses for unconditional benchmarks (e.g., WikiText-2), complementing the existing results for conditional settings.


- **Discussion of dependence and multiple comparisons**:
We included guidance on potential non-i.i.d. structures in language-model evaluation and discussed multiple pairwise comparisons with appropriate error-rate control.


- **Computational details and transparency**:
We reported implementation details, including computational details of likelihood evaluation and dataset processing.


- **Clarifications and minor revisions**:
We revised several sections for clarity, added discussions on the interpretation and limitations of statistical significance under the proposed metric, and addressed dataset-specific preprocessing details.


A detailed, point-by-point response to all reviewer comments has also been posted. We thank the reviewers for their constructive feedback, which helped improve both the clarity and the scope of the paper.

---

### Decision · Action_Editor_16oG · 2026-03-12

**Recommendation:** Accept as is

**Additional Comments:**

Please ensure that the final version fully incorporates the rebuttal clarification.

**Audience:**

Yes

**Audience Explanation:**

All reviewers agreed that principled evaluation of generative models with uncertainty quantification is an important and emerging topic of clear interest to the TMLR audience.

**Claims And Evidence:**

Yes

**Claims Explanation:**

Reviewers vp9U and AUbY already found the claims to be supported in the first round, and although reviewer ayqR initially raised concerns about the motivation and interpretation of uncertainty, these were clarified during the rebuttal and revision.